# High-throughput chemical proteomics workflow for profiling protein citrullination dynamics

Rebecca Meelker González[1], Sophia Laposchan [1], Erik Riedel[1], Anna Fürst[2], Naomi O'Sullivan [3,4], Wassim Gabriel [5], Mathias Wilhelm [5,6], Percy A. Knolle [2], Guillaume Médard [7], Bernhard Kuster [3,8] & Chien-Yun Lee [1] ✉

Citrullination is a post-translational modification implicated in autoimmune and inflammatory diseases, yet its low abundance and lack of effective enrichment tools have limited proteome-wide analysis. Here, we develop a robust chemical proteomics workflow with improved specificity and throughput. This method builds upon glyoxal-based derivatization and incorporates a cleavable biotin linker for efficient peptide enrichment, release, and identification via mass spectrometry. Benchmarking demonstrates a > 10-fold increase in the detection of citrullinated peptides at sub-0.1% abundance. Applying this workflow to primary human neutrophils, we successfully monitor dynamic regulation, quantifying dose-dependent activation and inhibition by the PAD4 inhibitor GSK484. Furthermore, stimulation with the fungal pathogen *Candida albicans* reveals a "core citrullinome" conserved across distinct stimuli. Notably, extensive citrullination of linker histone H1 and structural proteins like lamin B1 suggests broad remodeling of cell architecture during NET formation. This workflow enables proteome-wide mapping of citrullination sites and facilitates its study across diverse biological contexts.

Protein post-translational modifications (PTMs) are essential regulators of cellular processes such as gene expression, structural integrity, and immune responses[1,2]. Despite their biological significance, many PTMs remain challenging to study due to the lack of effective detection and analytical tools. Among these, protein citrullination has gained increasing attention for its roles in health and disease[3], yet remains underexplored.

Citrullination, catalyzed by peptidylarginine deiminases (PADs), converts the guanidyl group of arginine residues into a ureido group, resulting in a mass increase of 0.98 Da and the loss of positive charge[4].

This modification can profoundly alter protein structure and function, influencing apoptosis[5], gene regulation[6], structural support[7], and immune response[8]. A prominent example is histone hypercitrullination, which drives chromatin decondensation during neutrophil extracellular trap (NET) formation[9]. Aberrant citrullination is implicated in several diseases such as rheumatoid arthritis (RA)[4], neurodegenerative disorders[10], cancer[11], and viral infection[12]. Notably, anticitrullinated protein antibodies (ACPAs) serve as diagnostic tools for RA, highlighting the clinical relevance of citrullination and its potential as a therapeutic target[13].

[1]Young Investigator Group: Mass Spectrometry in Systems Neurosciences, School of Life Sciences, Technical University of Munich, Freising, Germany. [2]Institute of Molecular Immunology, School of Medicine and Health, Technical University of Munich, Munich, Germany. [3]Chair of Proteomics and Bioanalytics, School of Life Sciences, Technical University of Munich, Freising, Germany. [4]Institute of Pathology, School of Medicine, Technical University of Munich, Munich, Germany. [5]Computational Mass Spectrometry, School of Life Sciences, Technical University of Munich, Freising, Germany. [6]Munich Data Science Institute, Technical University of Munich, Garching, Germany. [7]Proteomics Core Facility, School of Science, National and Kapodistrian University of Athens, Athens, Greece. [8]German Cancer Consortium, Partner Site Munich, Munich, Germany. ✉e-mail: chienyun.lee@tum.de

Developing enrichment strategies coupled with mass spectrometry has been a turning point in PTM analysis, transforming our ability to study modifications such as phosphorylation at scale[2]. Citrullination, however, presents unique analytical challenges that make enrichment particularly critical. The 0.98 Da mass difference to arginine is identical to that of deamidation on Gln/Asn, complicating confident identification, particularly when multiple potential sites are present in a peptide. In addition, errors in assigning monoisotopic peptide mass can fall within set mass tolerances and increase the false positive rate during database searching[14,15]. Although advanced fragmentation techniques[16], search strategies[15,17], statistical models leveraging neutral loss patterns[16,17], and deep learning-based tools[18] have improved identification, detecting low-abundance citrullinated peptides in complex mixtures remains difficult. Deep proteomic profiling via orthogonal separation prior to MS measurement can enhance coverage[14,19], but is time- and resource-intensive, limiting scalability. As such, effective enrichment prior to MS remains essential for enabling sensitive and high-throughput citrullinome profiling.

Although antibody-based enrichment has proven effective for certain PTMs, such as phosphotyrosine and ubiquitination[20,21], no anti-pan-citrullination antibodies currently exist that can robustly and efficiently enrich citrullinated peptides[22]. As a result, chemical derivatization strategies have become the primary approach for citrullination enrichment[23–28]. These methods leverage the reactivity of citrulline's ureido group with glyoxals under acidic conditions[28] and typically use conjugation to agarose beads[23] or biotin moieties[25–27] for downstream capture. However, existing strategies suffer from several limitations, including poor peptide fragmentation[25], loss of site-specific information[26], harsh elution requirements for biotinylated peptides from streptavidin beads[27], and rely on in-house synthesized probes[27]. Critically, none are readily compatible with high-throughput workflows, posing a major barrier to broader implementation in citrullination proteomics.

In this work, we develop a high-throughput chemical workflow for global citrullinome profiling with high sensitivity and reproducibility to address current limitations (Fig. 1a). This workflow combines glyoxal-based derivatization of peptides using a clickable and cleavable linker, followed by biotin-based enrichment and on-bead cleavage of derivatized peptides for downstream analysis via mass spectrometry. We systematically optimize derivatization, enrichment, and database search strategies. All reagents are commercially available, and the workflow is validated in a 96-well format for streamlined scalability with standard LC-MS/MS pipelines. Using this workflow on the mouse brain expands citrullinome coverage, including proteins involved in myelin formation and synaptic vesicle transport, relative to a previous study[27]. In ionomycin-activated human neutrophils, dose-dependent profiling reveals dynamic citrullination during early neutrophil extracellular trap (NET) formation, with site-specific regulation across all core and linker histones. Notably, linker histone H1 shows pronounced citrullination, implicating its major role in NET biology. We also observe lamin B citrullination within direct DNA-binding region, suggesting its involvement in nuclear lamina destabilization, aligning with previous reports of lamina rupture during NET formation[29] and expanding citrullination's role beyond chromosome decondensation.

## Results

### A clickable-cleavable derivatization strategy for global profiling of citrullination

Biotin-conjugated chemical probes represent a primary approach for enriching citrullinated proteins and peptides from complex proteomes[25–27]. While probe designs vary, they all exploit glyoxals' reactivity with the citrulline ureido group under acidic conditions[28]. However, they face two key limitations: (1) the bulky derivatized moiety often impairs peptide fragmentation during tandem MS, reducing

identification and site-localization[25,26]; (2) probes showing improved fragmentation in tandem MS, e.g., a recent biotin thiol probe[27], still require harsh elution of derivatized peptides from streptavidin beads and are not commercially available.

To address these issues, we designed a two-step derivatization strategy compatible with tandem MS and large-scale citrullinome profiling using commercially available reagents (Fig. 1b; Supplementary Fig. 1a): 4-azido-phenyl glyoxal (APG), which reacts with citrulline and introduces a clickable azide handle (+158.01 Da), and Dde-PEG-biotin alkyne, a cleavable biotin with an alkyne moiety (+809.35 Da after click reaction)[30,31]. The final product after hydrazine cleavage leaves a + 213 Da mass tag compatible with MS detection.

We first developed the derivatization using the synthetic citrullinated peptide SAVRA[R(Cit)]SSVPGVR. Optimal derivatization was achieved under 50% TFA with an APG-to-peptide ratio up to 1000 (Supplementary Fig. 1b–c). MALDI-TOF MS confirmed the masses of the intermediate and final product at each step (Fig. 1c), including a + 131 Da shift after APG labeling due to $N_2$ loss during ionization[32] (Supplementary Fig. 1d), and a + 809 Da shift after click reaction, with characteristic PEG fragmentation[33] (Supplementary Fig. 1e). Hydrazine cleavage yielded the final +213 Da tag. No modification was observed in control peptides containing native arginine, confirming its specificity toward citrulline (Supplementary Fig. 1f). To test broader applicability, we applied the workflow to a pool of 185 synthetic citrullinated peptides (Cit Pool)[34] and matched arginine controls (Arg Pool), analyzed via LC-Orbitrap MS. MS Fragger open search[35] analysis showed a dominant +213 Da mass shift only in the Cit Pool (Fig. 1d; Supplementary Data 1), demonstrating high specificity across various peptides. Approximately 80% of peptides were recovered, with <1% residual signal from non-derivatized forms (Fig. 1e; Supplementary Data 2).

While modified peptides often generate diagnostic ions or neutral losses (NL) during fragmentation aiding in identification and localization[36], no such unique diagnostic ions were observed when examining the fragmentation patterns of derivatized peptides in HCD MS2 spectra (Supplementary Fig. 1g). However, fragment y-ions containing the derivatization showed neutral losses of −17 Da ($NH_3$), −28 Da ($N_2$), or −45 Da (combined) (Fig. 1f). These losses appeared in ~35–50% of modified fragment ions and ~18–30% of precursors (Fig. 1g; Supplementary Data 3). Although not universal, incorporating neutral losses into search scoring may aid in more accurate identification and localization from complex samples[14].

### Evaluation of a high-throughput enrichment workflow reveals robustness and reproducibility in complex proteomes

After validating efficient derivatization on synthetic peptides, we evaluated whether this strategy enables robust enrichment of citrullinated peptides in complex proteomes. To systematically assess sensitivity, efficiency, and reproducibility, we designed benchmark experiments under controlled and biologically relevant conditions. All steps were implemented in a 96-well plate format to support scalability (Fig. 2a).

We first assessed sensitivity and enrichment efficiency by spiking either a single citrullinated peptide or a pool of citrullinated peptides (Cit Pool) into HeLa tryptic digest, which has low levels of endogenous citrullination (Fig. 2a, *Exp. 1–2*). The single derivatized peptide was enriched and quantified down to 39.1 fmol/μg of a background proteome (Fig. 2b; Supplementary Data 4). Additionally, the recovery of Cit Pool was similar (~75–80%) with or without background (Fig. 1e, Fig. 2c; Supplementary Data 5). Enrichment boosted peptide intensity drastically: derivatized peptides accounted for ~50% of total peptide intensity after enrichment, compared to ~0.6% before enrichment, an 83-fold increase (Fig. 2d).

To mimic variable citrullination levels more representative of biological samples (Fig. 2a, *Exp. 3*), we performed in vitro citrullination of HeLa lysate using recombinant PAD4, followed by tryptic digestion

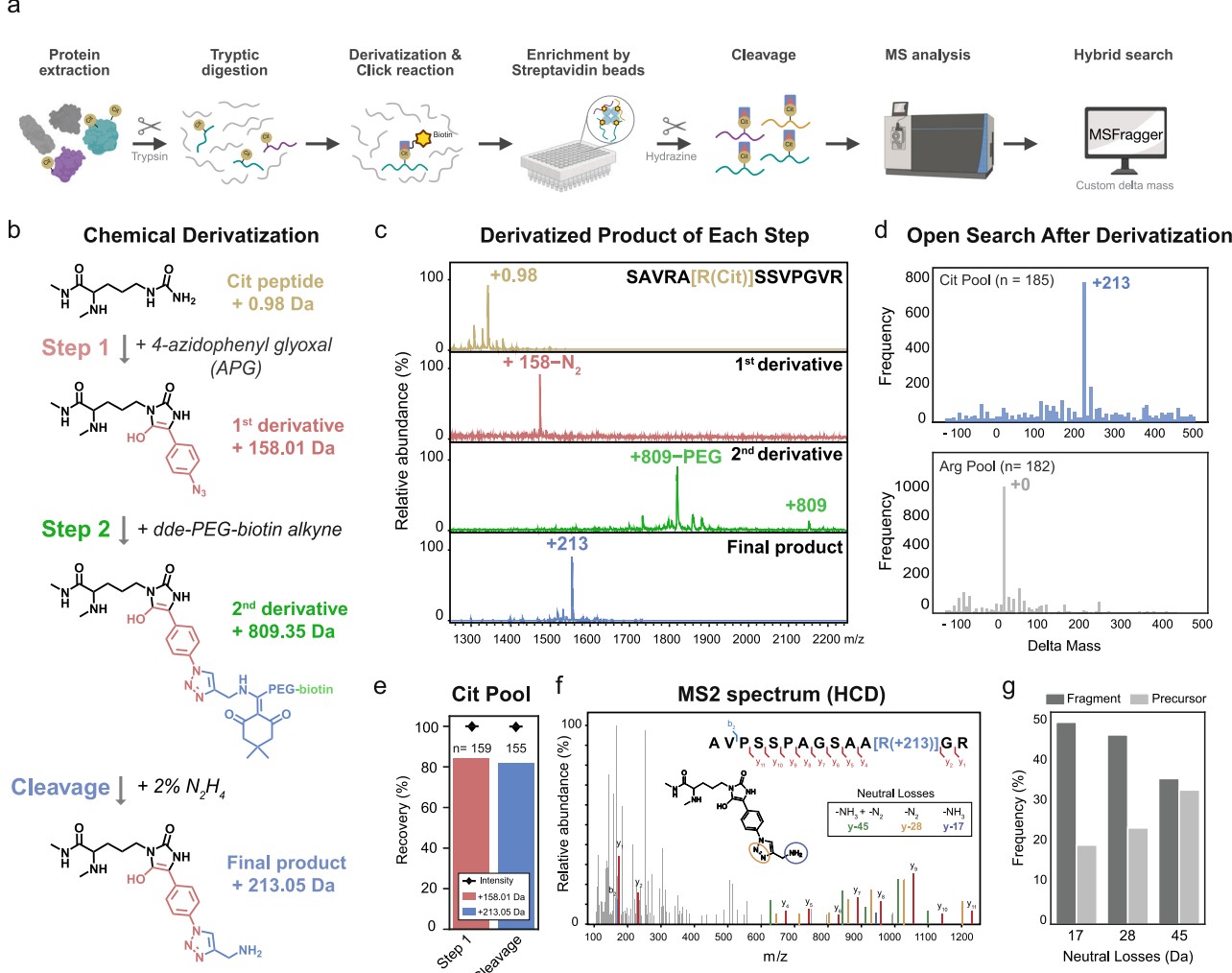

**Fig. 1 | A clickable-cleavable derivatization strategy for global profiling of citrullination. a** Overview of the workflow in complex biological systems. Cit, citrullination. Created in BioRender. Lee, C. (2026) BioRender.com/nkioa0e. **b** Stepwise derivatization of citrullinated peptides. Step 1: 4-azidophenylglyoxal (APG) selectively reacts with citrulline to form an azide-containing intermediate (1st derivative). Step 2: a copper-catalyzed azide-alkyne cycloaddition (CuAAC) introduces a biotin handle with a cleavable dde linker (2nd derivative). Final step: hydrazine treatment cleaves the linker, yielding a final derivative with a mass shift of 213.05 Da compared to the arginine-containing peptide (212.07 Da relative to the original citrullinated peptide). **c** MALDI-MS monitoring of each derivatization step using a synthetic citrulline-containing peptide (SAVRA[R(Cit)]SSVPGVR). **d** Mass

shift distribution from the open search of fully derivatized synthetic peptide pools containing citrulline (Cit Pool) or arginine (Arg Pool). **e** Peptide recovery and derivatization selectivity after Step 1 and full derivatization. Recovery is defined as the number of identified derivatized peptides divided by the total number of peptides in the Cit Pool. Intensity is calculated as the summed MS1 intensity of derivatized peptides divided by the total MS1 intensity of all identified peptides, including non-derivatized versions. Data are derived from a single synthetic peptide pool experiment. **f** Representative HCD MS2 spectrum of a derivatized peptide, showing diagnostic y-ions and characteristic neutral losses (−45 Da, −28 Da, −17 Da). **g** Frequency of observed neutral losses in derivatized fragment ions and precursors across Cit Pool.

and MS analysis. The resulting hypercitrullinated digest contained ~8–10% citrullinated peptides (by MS1 intensity). This digest was titrated into an untreated digest to produce a dilution series from 0.1–2% citrullination, covering physiological ranges[17]. Pre- and post-enrichment samples were analyzed to assess identification and quantification across this range.

We also tested whether the previously observed neutral losses in MS2 spectra (−17, −28, −45 Da; Fig. 1g) could improve identification. Using hybrid search from MSFragger[37], which incorporates neutral losses (−17, −28, −45 Da) from modified fragment ions into peptide scoring, we observed that peptides uniquely identified by hybrid search correlated more strongly with citrullination levels (Supplementary Fig. 2a–c; Supplementary Data 6), supporting its reliability for subsequent analyses. Across the titration series, enrichment drastically increased both identification and intensity of citrullinated peptides (Fig. 2e, f; Supplementary Data 6). Depending on the initial

citrullination level, we observed a 4- to 10-fold gain in peptide identifications and a 53- to 155-fold increase in intensity. Fold enrichment was greatest at lower citrullination levels. Notably, slight saturation occurred at 1–2% citrullination, which exceeds typical levels in vivo[17]. Motif analysis of enriched peptides revealed similar sequence preferences to unenriched samples, with Asp, Glu, and Gly enriched at the +1 position—consistent with previous findings[14,19] (Fig. 2g).

To assess technical reproducibility, we performed 12 replicates using samples with 0.25% citrullination (Fig. 2a, *Exp. 4*). On average, ~720 derivatized peptides were quantified per replicate, comprising ~45% of total peptide intensity (Fig. 2h–i; Supplementary Data 7). Roughly 400 peptides were consistently detected in ≥75% of replicates, with a median coefficient of variation of ~20% (Fig. 2j–k). These results demonstrate that the workflow is sensitive, reproducible, and well-suited for high-throughput citrullinome analysis, even in complex and low-abundance samples.

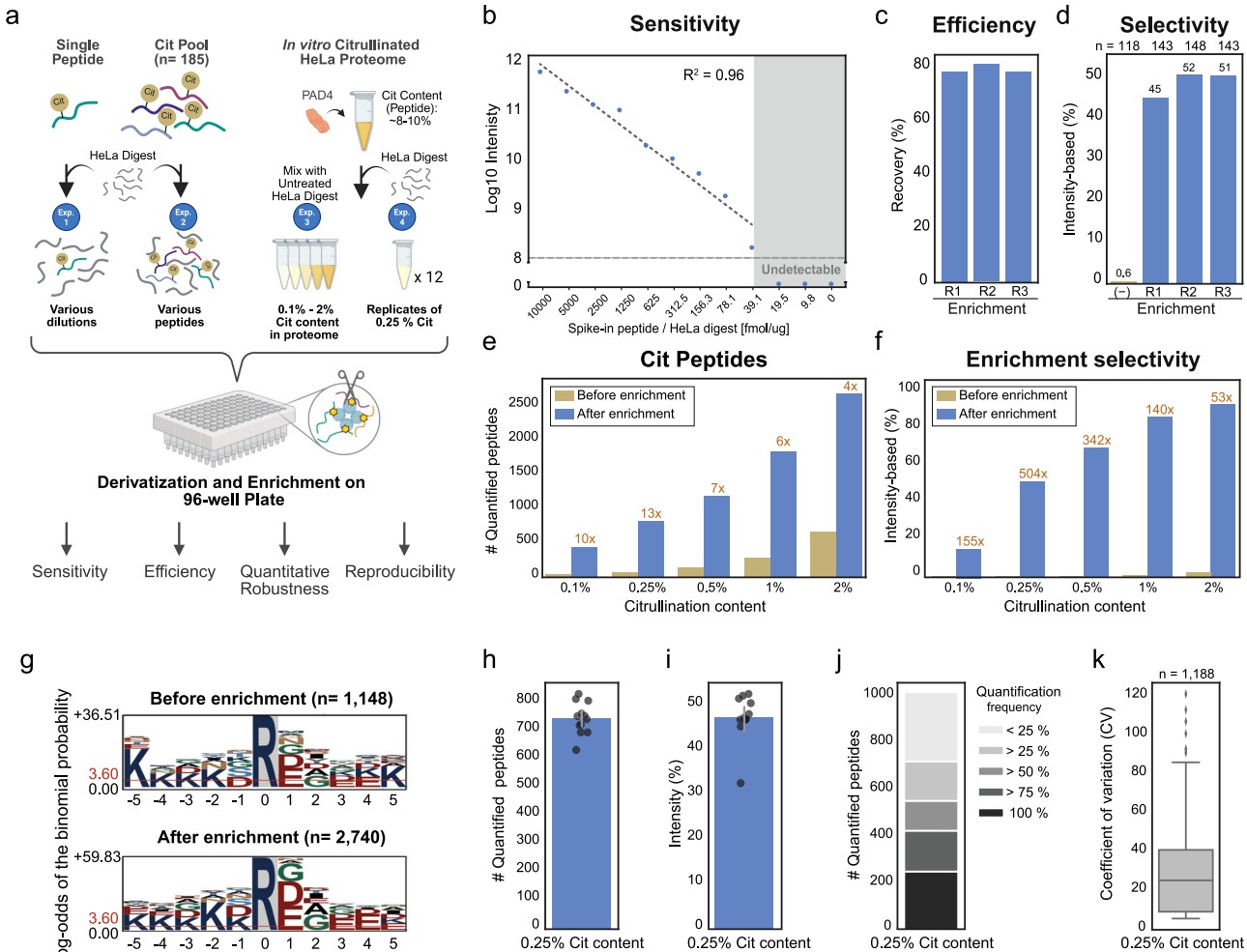

**Fig. 2 | High-throughput enrichment workflow demonstrates sensitivity, efficiency, and reproducibility in complex proteome background. a** Experimental setup of four benchmark experiments evaluating sensitivity (Exp. 1), enrichment efficiency (Exp. 2), quantitative robustness (Exp. 3), and reproducibility (Exp. 4). Created in BioRender. Lee, C. (2026) BioRender.com/nkioa0e. **b** Sensitivity assessment using a citrulline-containing synthetic peptide (SAVRA[R(Cit)] SSVPGVR) spiked into HeLa digest (Exp. 1). MS1 intensities correlate linearly with peptide input. **c** Enrichment efficiency using synthetic Cit pool spiked into HeLa digest (Exp. 2). Three technical replicates (R1–R3, $n = 3$) were analyzed. Recovery is defined as the number of identified derivatized peptides divided by the total number of peptides in the Cit Pool. **d** Intensity-based selectivity in Exp. 2, shown as the proportion of derivatized peptide MS1 intensity relative to total intensity of identified peptides, with a non-enriched control for comparison. The number of derivatized or citrullinated peptides is above the panel (n). **e** Quantitative

robustness was tested using a dilution series of PAD4-treated citrullinated proteome mixed with HeLa digest (Exp. 3). Samples with 0.1–2% citrullination content were analyzed before and after enrichment. Bars show the number of quantified citrullinated/derivatized peptides. The enrichment fold (orange) is shown above the bar and calculated as the ratio of peptide counts after versus before enrichment. **f** Corresponding intensity-based selectivity values based on peptide intensity from Exp. 3. **g** Sequence motif analysis of citrullinated peptides identified before and after enrichment (Exp. 3). **h–k** Reproducibility of enrichment assessed using $n = 12$ technical replicates (single biological sample processed in 12 independent wells; Exp. 4). Panels show peptide quantification count (**h**), selectivity (**i**), quantification frequency (**j**), and coefficient of variation (CV) across enriched peptides (**k**). Data in (**h** and **i**) are presented as mean values ± s.d. For the box plot (**k**), the center line represents the median, box limits represent the 25th and 75th percentiles, and whiskers extend to 1.5x the interquartile range (IQR).

## Citrullinome profiling of murine brain expands site coverage and reveals distinct patterns from a previous dataset

Citrullination is particularly abundant in the brain among both human[17] and mouse[27], yet its site-specific landscape and how it may vary across individuals or experimental workflows remains poorly defined. To explore this, we applied our workflow to three independent healthy brain samples of murine C57Bl/6 J mice and compared the resulting citrullinome to a previously published dataset generated using an alternative chemical probe (Shi et al.)[27]. While both studies identified ~200–300 citrullinated peptides per sample, Shi et al. reported a higher proportion of C-terminal citrullination—modifications on the C-terminal arginine of tryptic peptides—accounting for 62–65% of identifications versus 22–27% in our data (Fig. 3a; Supplementary Fig. 3a; Supplementary Data 8). Although the origin of C-terminal citrullination (biologically or technically) remains debated, cleavage

after citrulline by trypsin should occur at low rate due to its neutral charge[38]. Excluding these sites enhances confidence in subsequent comparative and quantitative analyses.

After removing C-terminal sites, our workflow identified 200–300 unique citrullination sites, 2–3 times more than the previous study[27] (Fig. 3b). However, only a tiny fraction of these sites, less than 4%, were shared between the two datasets (Fig. 3c). While this could reflect biological variation, differences in workflows (e.g., protein extraction, derivatization, instrumentation) may also play a role. Notably, our biological triplicates showed high internal consistency (Supplementary Fig. 3b), suggesting minimal intra-cohort variability. Protein-level overlap increased to 16.4% (Fig. 3d), with shared proteins enriched in myelin-associated components (Fig. 3e).

To further understand site specificity, we examined the sequence context of citrullinated sites across datasets to assess whether known

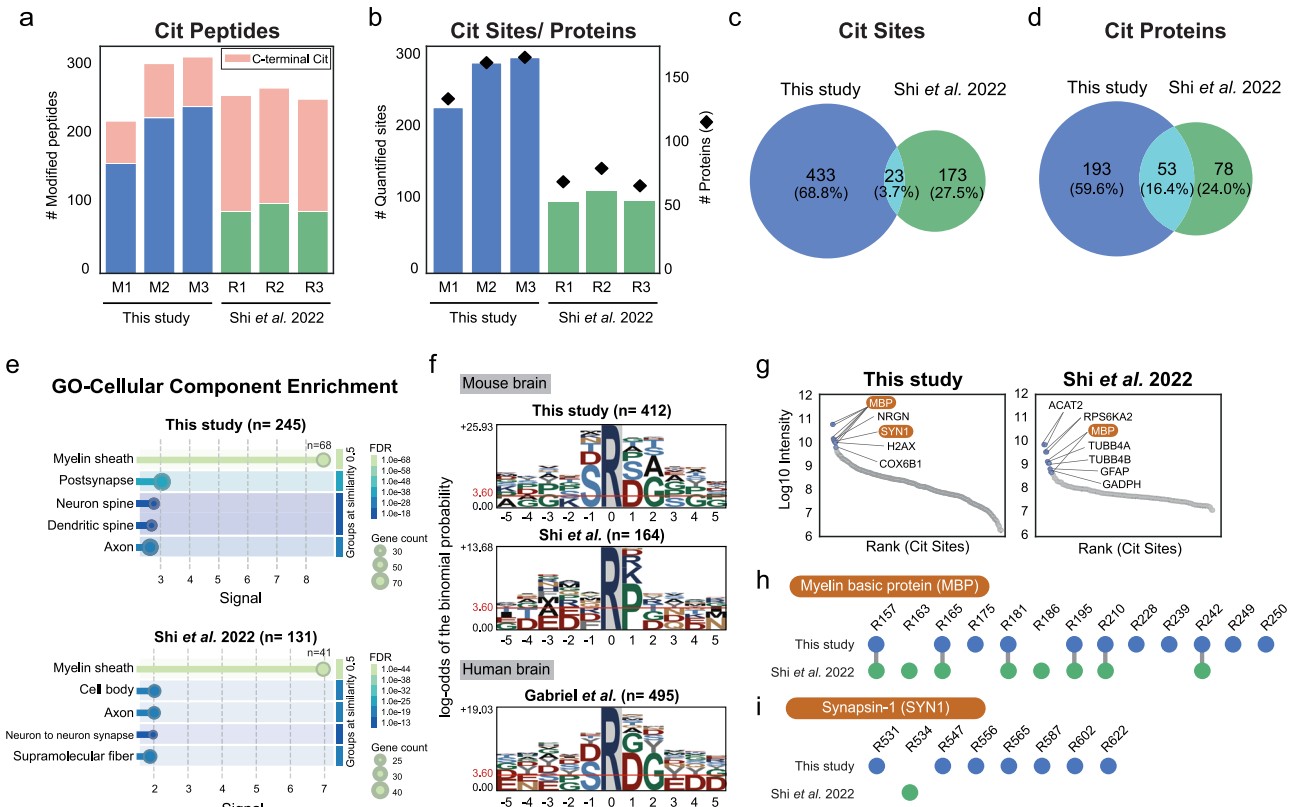

**Fig. 3 | Expanded citrullinome coverage in mouse brains reveals distinct profiles from a prior dataset. a** Number of citrullinated peptides identified across biological ($n = 3$ biologically independent animals; M1-3) in this study and technical replicates (R1-3) in Shi et al.[27]. Peptides with C-terminal citrullination are shown in light pink. **b** Unique citrullination sites after removing C-terminal sites. Overlap of identified citrullination sites (**c**) and proteins (**d**) between this study and Shi et al. **e** Gene Ontology (GO) enrichment of cellular components for citrullinated proteins using the mouse proteome as background (FDR < 0.05, similarity threshold = 0.5). **f** Motif comparison of citrullination sites from mouse brain (this study, Shi et al.) and human brain data (Gabriel et al.[18]). **g** Abundance rankings of citrullination sites from this study and Shi et al. The top 10 most abundant sites are annotated by gene name. Citrullination site mapping on myelin basic protein (MBP, **h**) and Synapsin-1 (SYN1, **i**) across datasets.

PAD substrate motifs had been preserved. Given conserved substrate recognition and active sites across human and mouse PADs[39], similar sequence preferences are expected. Our data recapitulated known PAD preferences from human brain[14,18], with additional enrichment for serine at the +1 position. In contrast, motifs from the previous study were less defined (Fig. 3f), potentially reflecting differences in enrichment or identification strategies.

To explore the functional context of citrullinated sites, we ranked them by abundance across both datasets. Six of our top ten sites—and two from the previous dataset—mapped to myelin basic protein (MBP), a well-established citrullinated target in the brain[40] (Fig. 3g, h). Both datasets captured six shared MBP sites; however, our study revealed five additional sites not previously reported, while two were unique to Shi et al. We also identified citrullination on other brain-enriched proteins, including seven sites on synapsin-1 (SYN1), a synaptic vesicle protein involved in neurotransmission (Fig. 3i). These findings suggest that citrullination in the brain may extend beyond myelin formation, potentially modulating broader aspects of neural function.

### Dose-dependent citrullinome profiling in activated human neutrophils reveals dynamic regulation of histones and structural proteins during NET formation

Neutrophils defend against pathogens via phagocytosis, degranulation, and formation of neutrophil extracellular traps (NETs)—web-like structures composed of chromatin, histones, and granule proteins released upon activation[41]. This process is driven by PAD4-mediated citrullination of histones[9,42]. While histone H3 citrullination is a widely

used NET marker, the full scope of protein citrullination during this process remains poorly characterized. This gap limits our understanding of its broader immunological roles, including the potential for citrullinated proteins to act as autoantigens[43].

To systematically define the NET-associated citrullinome, we profiled the citrullinome of primary human neutrophils stimulated with ionomycin, a calcium ionophore that activates PAD4 and induces NET formation[44] (Fig. 4a). Using a highly sensitive Orbitrap Astral MS, we quantified up to 1,700 citrullinated peptides on 580 proteins across ionomycin doses (0.01–3 μM) (Fig. 4b; Supplementary Fig. 4a; Supplementary Data 9). Citrullinated peptide intensity increased sharply between 0.3–3 μM—coinciding with NET-inducing conditions[45] (Supplementary Fig. 4b). Analysing dose-response data for statistical relevance using CurveCurator[46] revealed over 400 significantly regulated citrullination sites, including 384 up-regulated and 20 down-regulated sites (Supplementary Data 9), while overall protein abundance remained stable (Supplementary Fig. 4c; Supplementary Data 9), indicating stoichiometric regulation of citrullination.

We readily detected hallmark NET citrullination sites on histone H3 (R17, R26), as well as two additional regulated sites (R42, R49) (Fig. 4c). Notably, R17 and R26 citrullination co-occurred with H3 acetylation (K14, K18) and methylation (K27), suggesting potential PTM crosstalk that may facilitate chromatin decondensation. Beyond H3, we observed broad citrullination across all core histones (H2A, H2B, H3, H4), with prominent up-regulation of linker histone H1 variants (Fig. 4d). Mapping citrullination sites revealed N-terminal citrullination in H2A, H3, and H4, and widespread modification across H1

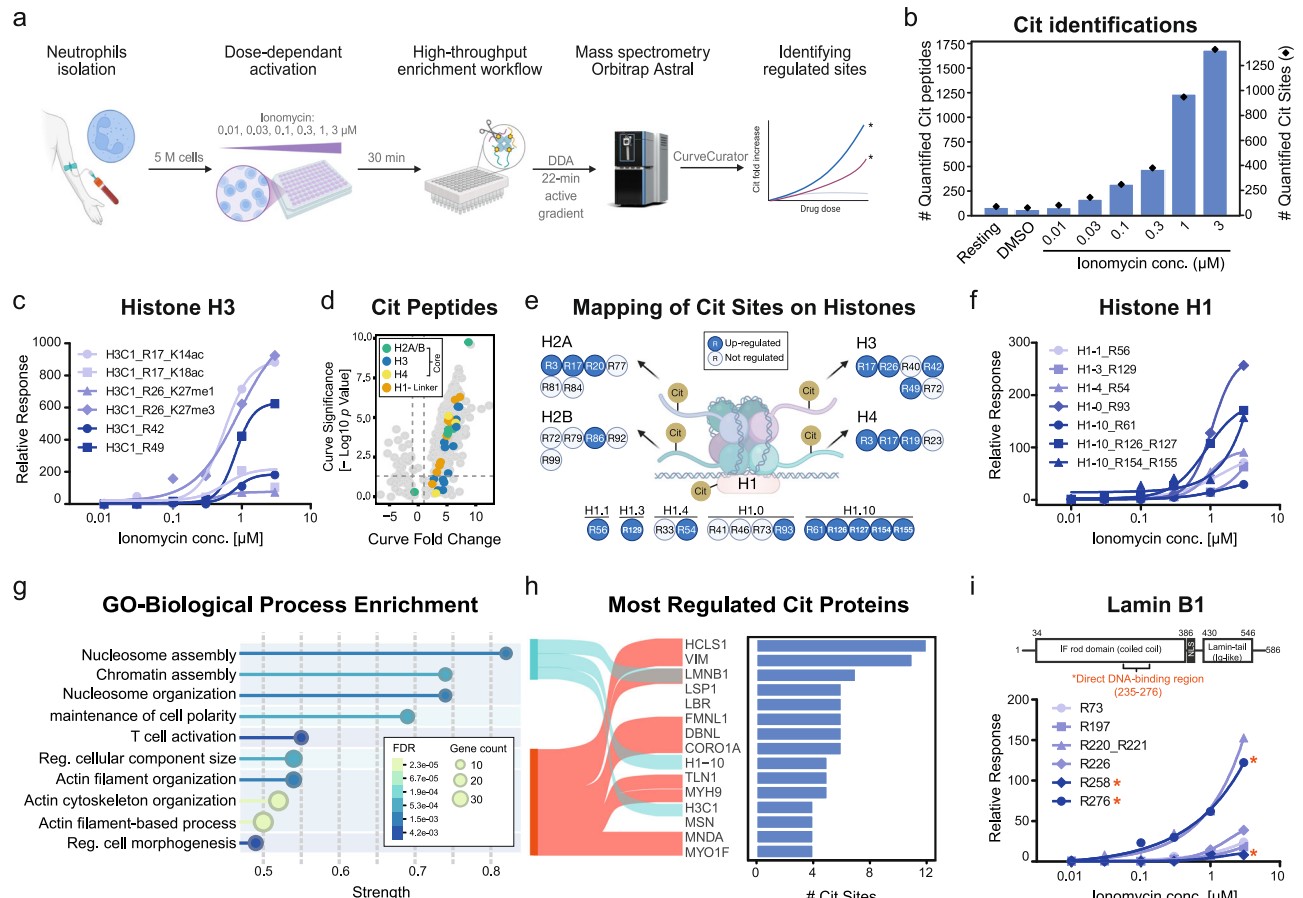

**Fig. 4 | Dose-dependent citrullinome profiling in human neutrophils reveals dynamic regulation of histone and structural protein citrullination during neutrophil extracellular trap formation. a** Profiling design using ionomycin-induced NET formation in human neutrophils (*n* = 1 biologically independent donor). Created in BioRender. Lee, C. (2026) BioRender.com/nkioa0e. **b** Number of unique citrullinated peptides (bars, left y-axis) and corresponding sites (dots, right y-axis) identified across resting, DMSO control, and ionomycin-treated cells. **c** Dose-dependent changes in citrullinated histone H3 peptides relative to DMSO control. **d** Volcano plot showing fold-change and significance of citrullinated peptides across doses, colored by histone subtype. Statistical significance was assessed using an F-test (comparing dose–response fit vs. flat line) with Benjamini-Hochberg FDR correction, as implemented in CurveCurator[46]. **e** Mapping of histone citrullination sites and their dynamics during activation. Classification of sites as up-regulated or not significantly changed (identified in at least one dose). Created in BioRender. Lee, C. (2026) BioRender.com/nkioa0e. **f** Dose-dependent changes of linker histone H1 citrullinated peptides. **g** Gene Ontology (GO) enrichment of biological processes based on significantly regulated citrullinated proteins using the neutrophils proteome as background (FDR < 0.05, similarity threshold = 0.5). **h** Sankey plot connecting enriched GO terms in nucleosome (turquoise) and cytoskeleton organization (red) to highly regulated citrullinated proteins (>3 up-regulated sites). **i** Schematic of Lamin B with domain structure and citrullination site dynamics. An asterisks mark sites within the direct DNA-binding region.

variants (Fig. 4e), including H1.4 at R54—a DNA-binding site implicated in autoimmunity[44,47]. H1.10 citrullination was especially highly up-regulated, suggesting an expanded role for H1 variants in chromatin decondensation (Fig. 4f). In contrast, down-regulated sites primarily mapped to secreted granule proteins (e.g., MPO, CTSG, LTF), likely reflecting protein secretion during neutrophil activation rather than active de-citrullination (Supplementary Data 9).

Gene ontology enrichment analysis of up-regulated proteins highlighted roles in nucleosome assembly, chromatin organization, and cytoskeletal remodeling—core processes for chromatin extrusion during NETosis[48] (Fig. 4g). Cellular component terms further indicated they are mainly in protein-DNA complexes and nucleosomes, consistent with the molecular functions of its binding to actin and DNA (Supplementary Fig. 4d-e). Strikingly, 10 of the top 15 most regulated proteins are involved in cytoskeleton organization (Fig. 4h), suggesting widespread remodeling of cell architecture via citrullination.

Among these, HCLS1 (HS1, Hematopoietic lineage cell-specific protein), an actin-regulatory protein, had 12 up-regulated citrullination sites (Fig. 4h)—10 of which localized to its N-terminal region overlapping Arp2/3 and F-actin binding domains (residues 27–212) (Supplementary Fig. 4 f). Given HCLS1's role in actin polymerization and

phagocytosis[49], citrullination in this region may disrupt cytoskeletal dynamics during NET formation. Similarly, lamin B1, a major component of the nuclear lamina, also emerged as a highly citrullinated protein, with seven up-regulated sites including two within its direct DNA-binding region (residues 242–276)[50] (Fig. 4i; Supplementary Fig. 4g). As lamin B1 anchors chromatin and maintains nuclear shape[51,52], citrullination may weaken chromatin-lamina interactions, promoting nuclear rupture as part of NET release[29].

To further investigate biophysical features of regulated citrullination, we examined their structural context using NetSurfP-3.0[53]. Compared to (non-modified) arginines, regulated citrullination sites were more frequently solvent-exposed and enriched in disordered or coil regions (~75% vs. ~60%) (Supplementary Fig. 4h-i). These features support citrullination's role as a regulatory PTM, preferentially targeting accessible and intrinsically disordered regions of proteins[54]. Given these characteristics and the established role of PAD4 in NET formation, we next examined the contribution of PAD4 activity to the observed citrullinome using the selective inhibitor GSK484.

Treatment with increasing concentrations of GSK484[55] (0.3–30 μM) resulted in a dose-dependent reduction in both citrullinated peptide intensities and site detection (Supplementary Fig. 5a–c), with a total of

1568 citrullination sites quantified across all conditions. Among these, over 1100 sites overlapped with the ionomycin-induced dataset (Supplementary Fig. 5d). Within this shared subset, 190 sites were significantly down-regulated by GSK484 treatment, defining a PAD4-dependent citrullinome. Notably, 103 of these sites were also up-regulated during ionomycin stimulation, establishing a core citrullinome directly modulated by PAD4 during neutrophil activation. We also identified 208 sites that were up-regulated by ionomycin but not significantly inhibited by GSK484, suggesting potential contributions from PAD2 activity. Interestingly, the majority of detected sites showed no significant regulation in either dataset, which may reflect lower abundance, stochastic detection, or regulation independent of PAD activity.

To estimate the inhibitory potency of GSK484, we examined the half-maximal effective concentration (pEC50) of regulated citrullination sites, revealing a median $EC_{50}$ of ~760 nM—approximately three-fold higher than the reported in vitro value[55] (Supplementary Fig. 5f). Among the GSK484-responsive substrates were canonical PAD4 targets such as histone H3 (R17, R26), as well as histone H1.10 (R127, R128, R155, R156) and lamin B1 (R197, R220, R221, R276)—the latter previously identified as a highly citrullinated protein during ionomycin stimulation (Supplementary Fig. 5g; Fig. 4f & i). Additional PAD4-dependent sites were found on nuclear RNA- and DNA-binding proteins involved in chromatin organization, including HP1BP3 and HNRNPU (Supplementary Fig. 5h), reinforcing the role of PAD4 in shaping the nuclear citrullinome during NET formation.

## Pathogenic stimulation with *Candida albicans* reveals a conserved NET-associated citrullinome

To further evaluate the biological relevance of our method under physiologically meaningful stimuli, we treated primary human neutrophils with heat-killed *Candida albicans* (HKCA)—a common opportunistic fungal pathogen known to trigger NET formation through PAD4-dependent mechanisms[55,56] (Fig. 5a). Across four biological replicates, we identified 268 citrullination sites, of which approximately 200 were consistently detected in at least two replicates (Fig. 5b). This represents a substantial increase compared to resting or serum-treated controls, which exhibited only ~40 sites, and supports PAD activation upon fungal challenge.

Although the total number of sites was lower than in ionomycin-treated cells, this is consistent with prior reports showing reduced histone H3 citrullination in *C. albicans*- versus calcium ionophore–stimulated neutrophils[56]. Importantly, when comparing the HKCA- and ionomycin-induced citrullinomes, we observed a huge overlap of 234 sites across 133 proteins (Fig. 5c, d). This suggests that, despite divergent upstream signaling, the downstream citrullination landscape during NET formation is largely conserved.

Gene ontology enrichment analysis of HKCA-induced citrullinated proteins revealed enrichment in biological processes such as nucleosome assembly, chromatin remodeling, and organelle organization (Fig. 5e)—highly similar to the processes observed in ionomycin-stimulated cells (Fig. 4g). Moreover, comparison of proteins with four or more citrullination sites confirmed extensive overlap between the two datasets, defining a robust, shared NET-associated citrullinome (Fig. 5f). Interestingly, several highly citrullinated proteins in the HKCA condition—such as ACTB, KRT1, and RPS6— while detected in the ionomycin dataset, were not significantly regulated. These may represent constitutively modified proteins present in basal neutrophils, rather than NET-specific citrullination events.

Together, these findings demonstrate that our enrichment platform captures a conserved and biologically meaningful citrullinome across distinct NET-inducing stimuli, reinforcing its utility for dissecting citrullination dynamics in both experimental and disease-relevant contexts.

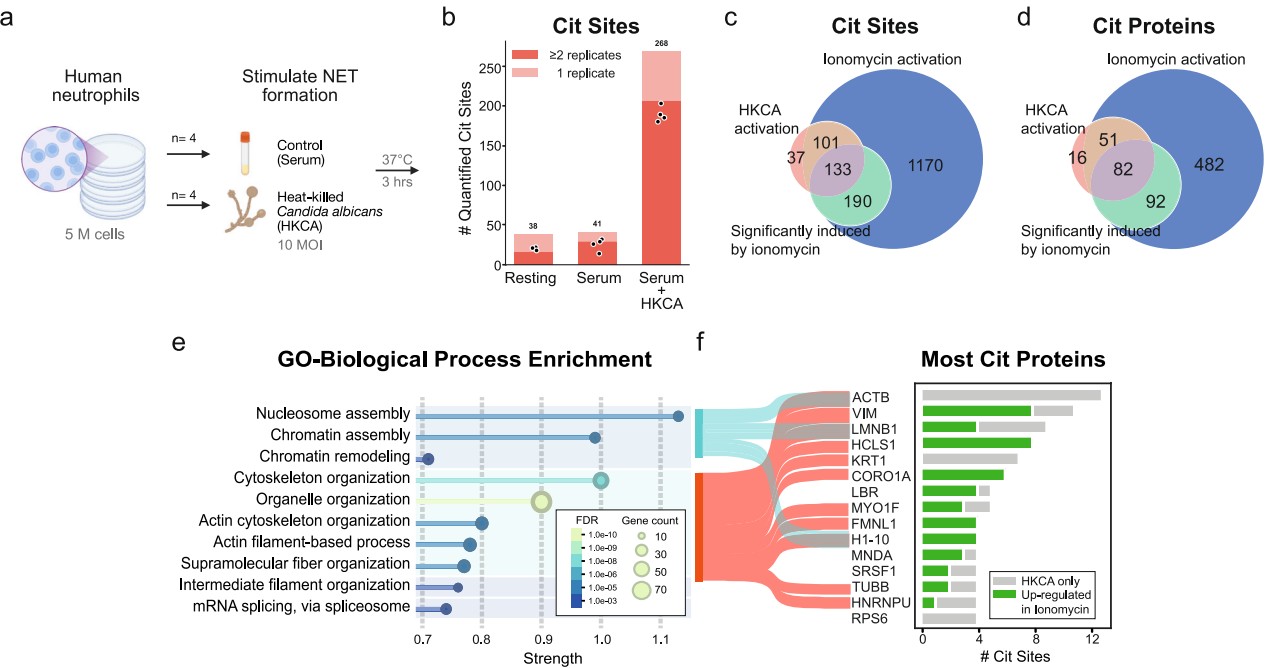

**Fig. 5 | Conserved core citrullinome in pathogen-induced NET formation.**
**a** Experimental workflow. Primary human neutrophils were stimulated with heat-killed Candida albicans (HKCA) or treated with serum control for 3 h ($n = 4$ independent sample preparations from a single donor). MOI, Multiplicity of Infection. Created in BioRender. Lee, C. (2026) BioRender.com/nkioa0e **b** Number of citrullination sites quantified in resting ($n = 3$), serum-control ($n = 4$), and HKCA-stimulated neutrophils ($n = 4$). Dots represent individual replicates. **c** Overlap of citrullination sites identified in HKCA-stimulated versus ionomycin-stimulated neutrophils (from Fig. 4). The intersection defines the conserved "core" citrullinome. **d** Overlap of citrullinated proteins between HKCA and ionomycin stimulation. **e** Gene Ontology (GO) biological process enrichment analysis for the shared citrullinated proteins. **f** Sankey diagram (left) and bar plot (right) visualizing the most heavily citrullinated proteins. Green bars indicate the number of sites shared with and up-regulated upon ionomycin stimulation; grey bars indicate sites detected upon HKCA stimulation.

## Discussion

Chemical derivatization has long held promise for the selective enrichment and identification of protein citrullination, but prior strategies were hindered by poor MS compatibility, inefficient peptide release, and limited scalability[23–27]. To overcome these challenges, we developed a proteomics workflow that uses clickable and cleavable derivatization for selective peptide labeling and clean release under mild conditions. The protocol is MS-compatible, uses commercially available reagents, and supports 96-well plate processing, enabling high-throughput and reproducible profiling. Across diverse sample types—including cell lysates, tissues, and primary immune cells—this method achieved >10-fold increases in citrullinated peptide detection and >150-fold increases in intensity, even at sub-0.1% abundance.

To maximize the identification of these derivatized peptides, we found that search engine choice is critical[57]. Our comparative analysis suggests that MSFragger, employing a hybrid search and labile mode[37], offered enhanced sensitivity in our hands compared to standard search strategies (Supplementary Fig. 6). This is likely because the derivatized citrulline moiety undergoes neutral loss during fragmentation; utilizing these diagnostic remnant ions allows for confident identification even when the intact modification mass is not fully preserved. As computational tools evolve, we anticipate that search algorithms will become increasingly optimized for such chemically complex modifications, further improving depth and confidence.

Application of our workflow to mouse brain tissue expanded citrullinome site coverage relative to a previous dataset[27]. While overlap at the site level was modest, our data reported strong reproducibility across biological replicates and identified conserved motifs shared with the human brain. The discrepancy between studies likely reflects both technical and biological factors such as mouse strain and age, extraction protocols, and MS acquisition strategies. Notably, while the brain exhibits the highest levels of protein citrullination among tissues[14,27], our understanding of its physiological roles and dynamics in between individuals remains limited—partly due to the lack of enrichment tools for consistent, large-scale profiling. For example, citrullination of myelin basic protein (MBP) has been shown to vary with age, with higher levels in younger individuals potentially contributing to looser binding between myelin sheath and axon, suggesting increased neuronal plasticity[58]. These highlight the importance of accounting for variables such as age, sex, and brain region in future studies. Standardized, high-throughput workflows will be essential to enable robust, cross-study comparisons and uncover brain citrullination's dynamic regulation in health and disease.

In primary human neutrophils, dose-dependent citrullinome profiling during NET formation revealed a broader and more dynamic citrullination landscape than previously appreciated[9,42]. Beyond calcium ionophore stimulation, we validated the physiological relevance of our findings using the fungal pathogen *Candida albicans*. We also attempted to induce NETosis using Rheumatoid Factor (RF)-immune complexes; however, consistent with previous reports[59], this stimulus failed to trigger robust NET formation in our primary neutrophil model. Importantly, while the total number of induced citrullination sites from *C. albicans* was lower than in the ionomycin-stimulated condition, we observed a high degree of overlap (~87%) between the ionomycin- and *C. albicans*-induced citrullinomes. This convergence defines a robust "core" citrullinome conserved across these two distinct stimuli. In addition to histone H3, we detected widespread modification of core and linker histones, cytoskeletal components, and nuclear envelope proteins. Notably, linker histone variants such as H1.10 emerged as prominent targets. While their role in chromatin organization is known[60], citrullination of H1.10 during NET formation remains poorly characterized. Our findings suggest that H1.10 citrullination may cooperate with H3 to promote overall chromatin decondensation. Citrullinated HCLS1 and lamin B1, involved in cytoskeletal remodeling and nuclear integrity, may contribute to nuclear rupture during NETosis, reinforcing the role of PADs as key drivers of this process[29,42,48].

We further validated the method's ability to quantify dynamic regulation—including down-regulation—through PAD4 inhibition. Pre-treatment with GSK484 resulted in a dose-dependent reduction of citrullination sites, identifying a subset of PAD4-dependent targets from basal modifications. Our findings align with recent deep-profiling efforts in neutrophil-like cells from Rebak et al.[19], sharing over 50% of identified sites (Supplementary Fig. 7). While Rebak et al. reported a larger citrullinome atlas using cultured cell lines (HL-60), extensive offline fractionation, and different analysis pipelines, our study focused on identifying high-confidence, functionally regulated sites in primary cells. By utilizing single-shot runs (22 min active gradient) with a dose-dependent design, our workflow captures the functional core citrullinome with high scalability. This throughput is crucial for processing the larger cohorts necessary to distinguish pathological signaling from the basal citrullinome.

Beyond structural remodeling, many citrullinated proteins identified here may act as autoantigens. For example, citrullinated R54 on histone H1, a target of autoantibodies in systemic autoimmunity[44], was among the regulated sites. In total, 19 citrullinated proteins in our dataset overlapped with entries in the human autoantigen database[61]. This convergence suggests that proteins released during excessive NET formation may promote autoantibody production under inflammatory conditions. These findings also highlight the potential of our workflow to uncover site-specific NET-derived citrullinated autoantigens and advance the understanding of autoimmune pathogenesis. From a structural perspective, regulated citrullination sites were enriched in disordered and solvent-exposed regions, supporting a model in which citrullination disrupts protein-protein or protein-DNA interactions[54]. This structural bias aligns with the role of citrullination in modulating chromatin organization and cytoskeletal dynamics, particularly during stress responses like NETosis.

While the presented workflow establishes a robust foundation, certain technical and computational avenues merit further optimization. First, regarding chemical specificity, we observed that glyoxal-based reagents can yield a minor side-reaction with arginine (+212.0700 Da), likely arising from the basic conditions used in SCX clean-up[62]. Although this signal is mass-distinguishable from the citrulline derivative (+213.0538 Da), it represents a background artifact that requires careful data management. Therefore, to ensure high-confidence identification, we recommend incorporating specific precautions into the analysis pipeline: strictly distinguishing the +212.07 Da artifact from the +213.05 Da citrulline shift and explicitly filtering out C-terminal modifications, where this side reaction predominantly occurs (Supplementary Fig. 8). This localization bias is inherent to the sample preparation, as tryptic digestion specifically cleaves after arginine residues, generating a vast population of peptides with C-terminal arginines that serve as substrates for this side reaction. Consequently, future workflows employing alternative proteases such as LysC could significantly mitigate this artifact at the source.

Second, the harsh acidic conditions (50% TFA) required for efficient derivatization may compromise other acid-labile PTMs. Consequently, while our method is highly specific for the citrullinome, it is not currently suitable for the simultaneous co-enrichment of modifications such as acetylation from the same aliquot. To address this in multi-PTM studies, this protocol should ideally be positioned as the final step of a serial enrichment workflow or performed in parallel.

On the computational front, accurately identifying and localizing sites on hyper-modified proteins, particularly histones, remains a challenge. We anticipate that incorporating machine learning–based rescoring into search algorithms will further enhance sensitivity and localization confidence in these complex regions. Finally, although the

workflow performs well across a range of citrullination levels, profiling very low-abundance samples still requires higher protein input, which can be limiting for scarce primary cells. Applying data-independent acquisition (DIA) represents a promising solution to lower input requirements, provided that high-quality spectral libraries tailored to specific experiment and sample type are developed.

In conclusion, this workflow opens opportunities to explore the functional landscape of this dynamic PTM across biological contexts. Its compatibility with standard proteomics platforms and adaptability to various sample types make it well-suited for time- or dose-resolved studies, clinical sample profiling, and functional studies in disease models. By expanding access to citrullinome analysis, this method opens avenues to unravel the functional impact of citrullination in both physiology and pathology.

## Methods

### Ethical statement
All experiments complied with relevant ethical regulations. Human neutrophils were isolated from peripheral blood donated by an author after obtaining written informed consent. The study adhered to the Declaration of Helsinki. In accordance with the guidelines of the Technical University of Munich, a formal ethics vote was not required for this basic research using human material ex vivo. Animal organs were obtained from mice sacrificed solely for organ removal. This procedure was performed in accordance with the guidelines of the Federation of Laboratory Animal Science Association (FELASA) and approved by the District Government of Upper Bavaria.

### Animals
Eight-week-old, male C57Bl/6 J mice ($n = 3$) were purchased from Charles River (Charles River Laboratories International, Inc., US). Animals were housed under specific pathogen-free conditions at Animal Core Facility of the School of Medicine and Health, TUM. Housing followed a 12 h light/12 h dark cycle with a temperature of $22 \pm 2\,^{\circ}\text{C}$ and a humidity of $55 \pm 10\%$. Animals were controlled daily.

### Synthetic peptides
Lyophilized synthetic peptides (GenScript Biotech) with sequences SAVRA[R(Cit)]SSVPGVR and SAVRARSSVPGVR were dissolved in 0.1% formic acid (FA) and further diluted to a final concentration of 10 pmol/μL with water before storage at −20 °C. A pool of 200 synthetic citrullinated peptides and a pool of their unmodified counterparts were described previously (JPT Peptide Technologies)[34]. These peptides were dissolved in 0.1% FA and 20% acetonitrile (ACN) to a concentration of 200 pmol/μL, followed by dilution to 20 pmol/μL in 0.1% FA.

### HeLa lysate, digestion, and desalting
HeLa cells (CCL-2; ATCC, cat#430641) were lysed in urea buffer (8 M Urea 40 mM Tris-HCl, pH 7.6). Dithiothreitol (DTT) was added to a final concentration of 10 mM, and the lysate was incubated at 30 °C for 30 min with shaking at 400 rpm. Subsequently, chloroacetamide (CAA) was added to 55 mM and incubated at room temperature for 30 min in the dark. The lysate was diluted with 50 mM ammonium bicarbonate and digested with trypsin (enzyme-to-protein ratio of 1:100) in two steps: 1 h at 30 °C followed by overnight incubation at 30 °C with shaking at 400 rpm. The samples were acidified to pH 1–2 with FA and desalted using 200 mg SepPak cartridges (Merck Millipore). Cartridges were equilibrated with 3 mL ACN, 3 mL 0.1% FA in 50% ACN, and twice with 3 mL 0.1% FA. Samples were applied by gravity, and the flow-through was reapplied once. The cartridges were washed twice with 3 mL 0.1% FA and eluted twice with 600 μL 0.1% FA in 60% ACN. For 50 mg SepPak cartridges, wash volumes were adjusted to 1 mL and elution volumes to 200 μL. Eluted samples were dried using a SpeedVac and stored at −80 °C.

### Spike-in experiments
Decreasing amounts of SAVRA[R(Cit)]SSVPGVR were spiked into 20 μg of digested HeLa peptides, ranging from 200 pmol to 0.2 pmol in half-step dilutions. A negative control without spiked peptide was included. Additionally, 1 nmol of the citrullinated peptide pool (CitPool) was spiked into 20 μg of HeLa peptides in triplicate. Samples were derivatized, enriched, and desalted for subsequent LC-MS/MS analysis.

### In vitro citrullinated HeLa lysate
HeLa lysates were cleaned and digested using the SP3 protocol[63]. A total of 200 μg of protein was precipitated with ACN on 10 μL of Sera-Mag A and B magnetic beads (GE Healthcare, cat. no. 45152105050250 and cat. no. 65152105050250), followed by washing with ethanol and ACN. For in vitro citrullination, 150 μL of PAD buffer (100 mM Tris, 2.5 mM DTT, 10 mM CaCl$_2$) was added to the beads, and PAD4 enzyme was added at a 1:50 enzyme-to-protein ratio. The reaction was incubated overnight at 37 °C with shaking at 800 rpm. PAD4 activity was halted by heating at 80 °C for 10 min. DTT (10 mM final concentration) was added and incubated at 37 °C for 1 h, followed by the addition of CAA (55 mM final concentration) and incubation in the dark for 30 min at room temperature. Samples were digested with trypsin (enzyme-to-protein ratio of 1:50) at 37 °C with shaking at 800 rpm overnight. Digests were centrifuged at $20{,}000 \times g$ for 5 min, and the supernatant was collected. Samples were desalted using 50 mg SepPak cartridges and dried in a SpeedVac. Citrullination content was assessed using LC-MS/MS. Aliquots were prepared by mixing citrullinated peptides with digested HeLa peptides to achieve samples with 0.1%, 0.25%, 0.5%, 1%, and 2% citrullination content (based on intensity) in a total of 100 μg of peptides.

### Mouse brain profiling
Eight-week-old male C57Bl6/J mice (Charles River Laboratories International, Inc., US) were euthanized utilizing isoflurane followed by cervical dislocation. Mice were perfused with PBS through the left heart ventricle and brain organs were harvested, washed in ammonium bicarbonate (50 mM, pH 7.8) and frozen in liquid nitrogen. Tissues were lysed using a lysis buffer containing 4% SDS and 40 mM Tris, followed by mechanical disruption with a TissueLyser II (QIAGEN, Germany), sonication using an R230 focused-ultrasound instrument (Covaris Ltd., UK, 300 s duration, 30 s on/off) and heat inactivation at 90 °C for 30 min. Protein quantification was measured using the BCA assay, following the manufacturer's protocol (Pierce, Thermo Fisher Scientific). Samples were digested and cleaned up using SP3 sample preparation as described above. Peptides were desalted with 50 mg SepPak cartridges and dried in a SpeedVac. A total amount of 400 μg of peptide per sample was enriched and analyzed using the described enrichment workflow.

### Neutrophil isolation, treatment, and profiling
Peripheral neutrophils were isolated from human whole blood of one healthy volunteer using the EasySep Direct Human Neutrophil Isolation Kit (STEMCELL Technologies) by immunomagnetic negative selection. Enrichment was carried out on 50 mL of blood as per the manufacturer's instructions. The cells were washed with HBSS (Hank's Balanced Salt Solution) without calcium and magnesium (Gibco, Thermo Fisher Scientific). Five million cells were kept as resting cells as control. The rest of the cells were centrifuged and suspended in HBSS with calcium and magnesium (Gibco, Thermo Fisher Scientific) (HBSS (+)). Five million cells in 1 mL HBSS of each treatment were treated with 1 μL of stock dilutions of ionomycin (Sigma-Aldrich; dissolved in DMSO; concentrations: 0.01, 0.03, 0.1, 0.3, 1 mM) and incubated at 37 °C for 30 min with gentle shaking at 700 rpm. For the inhibition experiment, the cells were treated with 1 μL of stock dilutions of GSK4844 (Sigma-Aldrich; dissolved in DMSO; concentrations: 0.3, 1, 3, 5, 10, 30 mM) at 37 °C for 30 min prior to ionomycin activation. Cells

were lysed using 2% SDS in 40 mM Tris-HCl (pH 7.6), followed by heat inactivation at 90 °C for 10 min. Samples were acidified to 1% TFA to hydrolyze DNA and neutralized by 3 M Tris buffer. Protein quantification was measured using the BCA assay, following the manufacturer's protocol (Pierce, Thermo Fisher Scientific). Each sample (200 μg of protein) was digested and cleaned up using SP3 sample preparation as described above. Peptides were desalted with 50 mg SepPak cartridges and dried in a SpeedVac. The peptides were enriched and analyzed using the described enrichment workflow. Dose-response data from citrullinated sequences were evaluated using CurveCurator[46].

### Neutrophil activation by heat-killed *Candida albicans* for proteomic analysis

Freshly isolated human neutrophils were prepared as described above and resuspended in HBSS containing calcium and magnesium (HBSS (+)) at a concentration of five million cells cells/mL. Heat-killed *Candida albicans* (HKCA; InvivoGen, $1 \times 10^9$ cells) was diluted in 1 mL of sterile, endotoxin-free water, and activation mixtures were prepared by combining the diluted HKCA with human serum (H4522, Merck) and HBSS (+). Control samples received serum[64] and HBSS (+) without HKCA. After gentle mixing at room temperature (400 rpm, 20 min), the activation mixture was added to neutrophil suspensions to achieve a multiplicity of infection (MOI) of 10:1 (HKCA:neutrophils). Cells were incubated at 37 °C for 3 h to induce activation. Following incubation, the medium was removed and cells were lysed in 2% SDS in 40 mM Tris-HCl (pH 7.6), heat-inactivated at 90 °C for 10 min, acidified with 1% TFA to hydrolyze DNA, and neutralized with 3 M Tris buffer as described above. Protein concentrations were determined by BCA assay (Pierce, Thermo Fisher Scientific), and 200 μg (protein) of each sample was processed using SP3-based digestion, peptide cleanup, and enrichment workflows as described for the ionomycin-treated samples.

### Enrichment of citrullinated peptides

Subsequent steps can be conducted in either 1.5–2 mL tubes or 96-well plate (Step1: BRAND® 96-well deep well plate Cat# BR701340, Merck; Step2: Nunc™ 96-Well Cat# 249944, ThermoFisher; Enrichment: 96-well filter plate Cat# SKU: 360021, Porvair Sciences).

Derivatization Step 1: Peptides were dissolved in 55 μL buffer containing 2.5 μL of 4-azidophenyl glyoxal (APG, 400 nmol/μL) (Apollo Scientific) in a 50% TFA/water solution. The reaction was incubated at 50 °C for 3 h, shaking at 1000 rpm. After incubation, the samples were diluted tenfold with water and desalting was performed using SCX StageTips. The samples were dried in a SpeedVac and stored at −20 °C.

Derivatization Step 2: The click buffer was prepared by combining 64 μL of water, 3.9 μL of Dde-biotin-alkyne (20 nmol/μL), 6.25 μL of THPTA (5 nmol/μL), 6.25 μL of CuSO4 (5 nmol/μL), and 31.25 μL of sodium ascorbate (50 nmol/μL). The dried samples from Step 1 were reconstituted using click buffer and incubated for 30 min at room temperature with shaking at 550 rpm. After incubation, the samples were acidified and desalted via SCX StageTips. The samples were dried in a SpeedVac and stored at −20 °C.

Enrichment and Cleavage: Derivatized peptides were enriched using Pierce High-Capacity Streptavidin Agarose beads (Thermo Fisher Scientific). After pre-wetting the wells with 1 ml of ethanol and PBS, 50 μL of beads were added to each well on a 96-well plate. The beads were washed twice with 500 μL PBS. The Step 2 sample was reconstituted with 0.2 % NP-40 in PBS buffer and incubated for 1 h with 400 rpm shaking followed by extensive washing (3 times 1 mL of 0.2 % NP-40 in PBS buffer, 3 times 1 mL of PBS buffer and 3 times 1 mL of water). Peptides were eluted with 100 μL of 2% hydrazine in water and incubated for 1 h at room temperature. Then the beads were washed twice with 100 μL of 2% hydrazine in water and the elutions were combined. The eluted peptides were acidified, desalted via SCX StageTips, dried, and stored at −20 °C for MS analysis.

### Strong cation exchange (SCX) Desalting

For peptide quantities up to 400 μg, StageTips were constructed using 300 μL tips packed with 4 SCX disks (2.8 mm diameter; IVA Analysentechnik GmbH). The tips were equilibrated with 300 μL ACN followed by 300 μL 0.1% FA via centrifugation at $250 \times g$ for 1 min. Acidified samples were loaded, reapplied once, and washed three times with 300 μL 0.1% FA in 50% ACN. Peptides were eluted with 200 μL of 2% NH₃ in 40% ACN. For smaller sample volumes or enriched synthetic peptides, smaller StageTips with 200 μL tips and 3 SCX disks (0.6 mm diameter) were used, with reduced washing and elution volumes.

### LC-MS/MS analysis

Orbitrap Fusion Lumos: Samples (synthetic peptides, in vitro citrullinated HeLa digest, and mouse brain tissues) were dissolved in 0.1% formic acid (FA) and analyzed using an Orbitrap Fusion Lumos mass spectrometer (Thermo Fisher Scientific) coupled to an UltiMate 3000 HPLC system. Sample injection was performed at a drawing speed of 0.5 μL/s. Peptides were initially loaded onto a trap column (ReproSil-pur C18-AQ, 5 μm, 20 mm × 75 μm, self-packed; Dr. Maisch) at a flow rate of 5 μL/min using solvent A (0.1% FA in water). Separation was performed on an analytical column (ReproSil Gold C18-AQ, 3 μm, 50 cm × 75 μm, self-packed; Dr. Maisch) at a flow rate of 300 nL/min. A linear gradient from 4% to 32% of solvent B (0.1% FA and 5% DMSO in ACN) was applied over 50 min (extended to 100 min for mouse brain samples). MS data acquisition was performed in data-dependent acquisition (DDA) mode with a cycle time of 2 seconds. MS1 spectra were acquired in the Orbitrap over a mass range of 360–1300 m/z at a resolution of 60,000, with an AGC target of 100% and a maximum injection time (maxIT) of 118 ms. Precursor ions with intensities above 2.5e4 and charge states of 2–6 were selected for MS2 analysis. Dynamic exclusion was set to 30 s. Fragmentation was performed using higher-energy collisional dissociation (HCD) at a normalized collision energy (NCE) of 35%. MS2 spectra were acquired at a resolution of 30,000 with an isolation window of 1.3 m/z, an AGC target of 150%, and a maxIT of 54 ms.

Orbitrap Astral: Neutrophil samples were dissolved in 0.1% FA and analyzed using a Vanquish Neo UHPLC system (Thermo Fisher Scientific) coupled to an Orbitrap Astral mass spectrometer (Thermo Fisher Scientific) with a Nanospray Flex ion source. The LC was operated in trap-and-elute mode using backward flush. Peptides were trapped on a PepMap Neo Trap Cartridge (300 μm × 5 mm) and separated using an IonOptics Aurora Rapid analytical column (75 μm × 8 cm) maintained at 50 °C. Mobile phase A was 0.1% FA in water and mobile phase B was 0.1% FA in ACN. The gradient, in total 27 mins (active gradient 22 mins), proceeded as follows: 2% to 6% B over 0.5 min at 750 nL/min, 6% to 8% B over 2 min while decreasing flow to 300 nL/min, then 8% to 32% B over 19 min. The column was washed with 80% B at 750 nL/min for 2 min, followed by equilibration at 2% B. MS data acquisition was performed in positive ion mode using DDA. MS1 scans were acquired in the Orbitrap at 240,000 resolution, every 0.6 s, over a scan range of 360–1,300 m/z. The AGC target was 300% (3e6 charges) with a maxIT of 3 ms. Precursor ions were fragmented using HCD at an NCE of 26%. MS2 scans were acquired in the Astral analyzer over a scan range of 100–2000 m/z using a 1.2 Th isolation window, an AGC target of 300% (3e4 charges), and a maxIT of 5 ms. MS2 selection included charge states 2–6 and undetermined, with dynamic exclusion set to 5 s and monoisotopic precursor selection (MIPS) enabled for peptides.

### Database search

Raw data files were analyzed using FragPipe v22 (MSFragger v4.1, IonQuant v1.10.27) against all canonical protein sequences as annotated in the Swiss-Prot reference database of human (20,435 entries, downloaded 28 June 2024) or mouse (17,212 entries, downloaded 24 July 2024). Strict trypsin was specified as the proteolytic enzyme, and

up to three missed cleavage sites were allowed. All the searches were applied with 1% FDR threshold on PSM level using PeptideProphet unless specified elsewhere.

Open database searches on synthetic peptide pools employed default open search settings[35]. Closed database searches on synthetic citrullinated peptide pool were conducted using default settings with additional variable modifications: deamidation of asparagine/glutamine (Δ mass 0.9840), citrullination (Δ mass 0.9840), first derivative (Δ mass 158.0116, $C_8H_2O_2N_2$), and the final derivative of arginine (Δ mass 213.0538, $C_{11}H_7O_2N_3$) were set as variable modifications, allowing up to two occurrences per peptide. Database searches on hydrolysis test dataset were conducted using default settings with nonspecific digestion.

Citrullinome of HeLa dilutions, mouse brain, and neutrophils were analyzed using hybrid search in MSFragger[37]. Hybrid searches were performed by modifying the default search settings in the MSFragger tab using a detailed mass offset file (#Mass: 213.0538, Fragment remainder ions: 168.0211, 185.0477, 196.0273) and enabling the labile search mode. Under the open search options, report mass offset as variable modification by removing the delta mass was selected and Localize Mass Shift (LOS) was enabled. Variable modifications including asparagine/glutamine deamidation (Δ mass 0.9840), and the final derivative of arginine (Δ mass 213.0538) (HeLa, mouse brain, and neutrophils) were permitted up to two occurrences per peptide. Additionally, for the neutrophil dataset, lysine/arginine methylation (Δ mass 14.0156), lysine/arginine dimethylation (Δ mass 28.0313), lysine trimethylation (Δ mass 42.0469), and lysine acetylation (Δ mass 42.0106) were permitted as variable modifications, with up to one occurrence per peptide. C-terminal derivatization was excluded for the down-stream analysis.

Global proteome of HeLa dilutions and neutrophils were analyzed using default closed search settings with additional variable modifications: deamidation of asparagine/glutamine (Δ mass 0.9840) and citrullination (Δ mass 0.9840) with up to two occurrences per peptide allowed and up to 3 miscleavages. For HeLa dilutions dataset, no FDR (100% FDR) was applied. The search results of citrullination identifications were further analyzed using Prosit-Cit at 1% FDR (PSM) for better precision[18]. For neutrophils dataset, additional variable modifications were added including: lysine/arginine methylation (Δ mass 14.0156), lysine/arginine dimethylation (Δ mass 28.0313), lysine trimethylation (Δ mass 42.0469), and lysine acetylation (Δ mass 42.0106) with up to one occurrence per peptide. C-terminal citrullination was excluded for the down-stream analysis.

## MALDI-TOF analysis

For MALDI-TOF experiments, 800 pmol of the synthetic peptide SAVRA[R(Cit)]SSVPGVR dissolved in 0.1% FA underwent derivatization following aforementioned enrichment workflow with below modifications. The peptide was enriched by incubating the sample with 10 μL of Streptavidin Mag Sepharose (Cytiva Life Sciences) at room temperature for 1 h. A StageTip was constructed on a 10 μL pipette tip using one C18 disk (0.5 mm diameter). The tip was conditioned with 20 μL of ACN and 0.1% FA. Half of the reaction mixture was loaded onto the StageTip, which was washed with 20 μL of 0.1% FA before eluting directly onto a MALDI target using 5 μL of α-Cyano-4-hydroxycinnamic acid (HCCA) matrix. Spectra were acquired using a MALDI ultrafleXtreme mass spectrometer (Bruker Daltonik). Data processing and visualization were performed in FlexAnalysis software (Bruker Daltonik) employing baseline subtraction and smoothing algorithms.

## Data analysis

Diagnostic ions exploration: Accumulated MS2 spectra were generated from spiked Cit Pool into HeLa samples using a custom Python script. Briefly, MS2 spectra corresponding to derivatized and

unmodified peptides were extracted from the psm output file from the Fragpipe search. The m/z and intensity values were taken from the corresponding mgf file. The intensities of both across m/z 100 to 400 were aggregated within tolerance windows of 0.0001 Da.

MS2 spectra visualization and annotation: Interactive peptide spectral annotator was used to visualize and export annotated MS2 spectra[65].

Annotation of neutral loss in MS2 spectra: To identify and count neutral loss-associated fragment ions in a given PSM, theoretical y- and b-ion masses were first computed for the peptide sequence using residue-specific molecular weights[66]. Briefly, for ions that can carry neutral losses adjusted masses were derived by subtracting the respective neutral loss mass from the theoretical y- and b-ion masses. The same procedure was applied to precursor ions, where expected neutral losses were subtracted from the intact precursor mass to generate adjusted target masses. Experimental spectrum was then searched for peaks matching these adjusted masses, with tolerances set according to the mass analyzer: ±20 ppm for high-resolution Fourier-transform mass spectrometers (FTMS). Matches were counted if their m/z values fell within the defined tolerance window.

Motif Analysis: Potential substrate motifs surrounding identified citrullination sites were analyzed using pLOGO[67]. Sequences spanning −5 to +5 residues around the citrullination sites were extracted and compared with their respective background frequencies in the human proteome. Significantly up-regulated motifs ($p = 0.05$, Bonferroni corrected) were visualized.

Gene ontology enrichment analysis: This analysis was performed using functional enrichment analysis in STRING database[68]. Gene names of citrullinated proteins with up-regulated sites in neutrophils dataset were annotated with GO terms. Enrichment was performed and visualized using human proteome as background with following filters: FDR <= 0.05, group terms with similarity >=0.5, strength >= 0.01, minimum count in network of 2. Top 10 terms were shown.

Secondary structure prediction: Secondary structure prediction was carried out using the NetSurfP-3.0 web server (DTU Health Tech, Denmark)[53]. Canonical fasta sequences of all proteins with a citrullination site found in the neutrophil data set were retrieved from Uniprot using the ID mapping tool to human UniProtKB/Swiss-Prot database (downloaded 13 Jan 2025). Fasta sequences were then submitted to the NetSurfP-3.0 online tool and the output downloaded in csv format. Protein Q9NZV6 (gene name MSRB1) had to be excluded from structure prediction due to the presence of a non-canonical amino acid (selenocysteine U) in its sequence which is not supported by the prediction tool. The resulting prediction data was filtered for arginine residues, and the citrullination sites found in the MS data mapped to their corresponding predicted structures. Non-modified arginine residues among the proteins carrying citrullination sites served as background for secondary structure and the relative solvent accessibility (RSA) comparisons.

## Reporting summary

Further information on research design is available in the Nature Portfolio Reporting Summary linked to this article.

## Data availability

The mass spectrometry proteomics data have been deposited to the ProteomeXchange Consortium (http://proteomecentral.proteomexchange.org)[69] via the MassIVE partner repository with the dataset identifier MSV000097617 [https://massive.ucsd.edu/ProteoSAFe/dataset.jsp?accession= MSV000097617]. All processed data generated in this study are provided in the Supplementary Information, Supplementary Data, and Source Data file. Source data are provided with this paper.

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

## Acknowledgements

The authors would like to thank Dr. Christina Ludwig, Dr. Andreas Zellner, Dr. Polina Prokofeva, Ms. Susanne Wudy, Mr. Genc Haljiti, Prof. Dr. Thomas Skurk, Dr. Kurt Rack, and members from the Bavarian Center for Biomolecular Mass Spectrometry (BayBioMS) and the Chair of Proteomics and Bioanalytics for their valuable assistance and insightful discussions. This work was funded by the German Federal Ministry of Education and Research: FKZ031L0215 (YIG-SysNS; C.Y.L.); FKZ161L0214A (CLINSPECT-M; B.K.); FKZ03LW0243K (CLINSPECT-M-2; B.K.) and ERC Starting Grant (grant number 101077037; M.W.). The Orbitrap Fusion Lumos and Orbitrap Astral mass spectrometers used in this study were funded in part by the German Research Foundation (DFG-INST 95/1436-1 FUGG & DFG-INST 95/1859-1 FUGG). The graphical illustrations were created with BioRender.com.

## Author contributions

G.M., B.K., and C.Y.L. conceptualized the study. R.M.G. and C.Y.L. designed experiments. R.M.G. performed experiments on development of the methodology. R.M.G., S.L., E.R., A.F., and N.O. performed experiments on biological samples. R.M.G., S.L., W.G., and C.Y.L. analyzed the data. M.W., P.A.K., and B.K. provided analysis tools, animal samples, and instrumentation. R.M.G. and C.Y.L. wrote the manuscript with input from all authors. All authors reviewed and edited the manuscript.

## Funding

## Competing interests

B.K. and M.W. are non-operational co-founders and shareholders of MSAID. All other authors declare no competing interests.
