## [Transparent Peer Review file · Nature Communications]

High-throughput chemical proteomics workflow for profiling protein citrullination dynamics

Corresponding Author: Dr Chien-Yun Lee

Version 0:

Reviewer comments:

Reviewer #1

(Remarks to the Author)
General

Meelker González et al. present a proteomics project in which they aim to enable the systemic study of citrullinated peptides via mass spectrometry, with the focus of specifically enriching this modification that has otherwise mainly been studied in a non-enriched (total proteome) context. Such an approach would allow the mass spectrometer to spend more time resolving citrullinated peptides, rather than irrelevant background peptides. In turn, this would enable the study of citrullination in more biologically relevant contexts, and at a scale that would be more affordable to many labs.

Current enrichment approaches for citrullinated peptides often lack specificity, are chemically quite harsh, or result in a purified sample that is not suitable for MS analysis. The authors sought to work around this via a two-step derivatization and enrichment strategy, ultimately leaving a 213 Da mass remnant for each citrulline. They first validate this strategy, benchmark it in a setting with synthetic and spiked-in hyper-citrullinated HeLa, and finally test out the method in mouse brain as well as human primary neutrophils treated with a gradient of ionomycin.

The manuscript is well written and logically structured. The authors clearly explain the methodology and prove its efficacy in a variety of settings, giving equal importance to technical validation and application of their methodology to relevant biology. The strategy would be applicable by other labs, and potentially could be further refined in the future to enhance systems-wide profiling of citrullination by MS.

I have a few queries regarding some of the MS data processing and quality-control throughout the manuscript. Overall, if these points are adequately addressed, I would recommend this manuscript for publication in Nature Communications.

Specific points

The authors define and search their mass remnants as “first derivative (Δ mass 158.0116), and the 509 final derivative of arginine (Δ mass 213.0538)”. In addition to mass deltas, the authors should provide the atomic composition of the derivatives relative to arginine, for the sake of clarity and so that this modification may be defined in other search engines. If I am correct, the atomic composition of the +213.05 would be C11 H7 N3 O2?

As the authors are using data-dependent acquisition and are trying to pinpoint the location of citrullination within the peptides, correct localization is of high importance. The authors also mention the importance of localization throughout the manuscript, and in the methods describe that “Localize Mass Shift (LOS) was enabled” within the hybrid search of MSFragger. How exactly does the localization in MSFragger work? I was able to find a localization probability in some of the MSFragger output, which seems to either be “0” or “1”. Why are there no probabilities in between? While localizing a relatively large (213 Da) and stable PTM should not be too challenging, it would be beneficial if the authors could make an estimation/visualization of the distribution of localization confidence for the citrullinated peptides, especially since they (rightfully) choose to exclude peptide C-terminal citrullination.

The authors should provide an estimate or distribution of overall spectral quality for all complex datasets (figures 2, 3, 4). The Astral data (Figure 4) seems to be of high spectral quality, via manual inspection of the RAW data and a quick search via MaxQuant. However, based on a manual investigation of some of the MS/MS spectra within e.g. the RAW data generated for Figure 3 (YIG_314_L001_105_002_RMG, YIG_314_L001_105_005_RMG, and YIG_314_L001_105_008_RMG) using a Fusion Lumos, the quality of these MS/MS spectra is overall not very high. Analyzing these files with MaxQuant yields a relatively low MS/MS identification rate (~7%), with average spectral scores of ~83 for all identified peptides, and only approximately 30% of peptides exceed an Andromeda score of 100 (which is generally a minimum for a spectrum that would be manually annotatable). For the +213 Da modified peptides, which appear to score lower than unmodified background peptides, only about 10% of spectra appear to be in excess of 100 Andromeda score. Based on a (quick and perhaps sub-optimal) MaxQuant search of the 3 mouse RAW files I was able to find ~100 unique +213 Da sites, whereas the authors find ~400 using MSFragger (in both cases excluding peptide C-terminal site assignments). Perhaps MSFragger (in combination with post-processing using PeptideProphet) is more efficient at identifying this modification compared to MaxQuant, but still the difference seems relatively large. Using an alternative search engine on the data and being able to provide numbers in the same ballpark as MSFragger, which could be included as a technical note, would strengthen the claims made by the authors.

In relation to the two points above, and since we are looking at DDA data for modified peptide sequences, the authors should provide fully annotated MS/MS spectra for all citrullinated peptides identified in the manuscript, stratified per experiment. Currently there is only one annotated spectrum in Figure 1F, which is missing virtually the entire b-ion series. Providing annotated spectra (e.g. online via MS-Viewer, or batch-exported from any DDA search engine) would increase transparency and simultaneously demonstrate sufficient spectral quality to facilitate accurate identification and localization of citrullination within the peptides. Without the ability to manually inspect annotated spectra, and perhaps also an orthogonal data search using a different software, it would be difficult to ascertain the spectral quality and rule out over-interpretation of the data.

The authors derivatize citrullination, and search for the 213 Da derivative on Arginine residues. Is it possible for this derivative to exist on other amino acid residues? While this is to an extent addressed in Figure 1, it would be important for the authors to perform an *in silico* entrapment experiment, by searching a subset of the data generated from complex samples (e.g. from Figure 2, 3, or 4) and allowing the 213 Da mass remnant to be assigned to e.g. glutamine and asparagine residues in addition to arginine residues (or ideally allow it on any residue to perform a true localization experiment, but this may be computationally challenging). This would give an indication of the overall false discovery rate of the author's methodology when applied to complex samples – either because of non-specific chemical derivatization (i.e. *in vitro*), or in relation to the accuracy of the computational localization (i.e. *in silico*).

In their final experiment (Figure 4), the authors enrich and identify citrullination from primary human neutrophils stimulated with ionomycin, which is a calcium ionophore. Although briefly mentioned in the introduction, a recent landmark publication by Rebak et al. (published in *Nat. Struct. Mol. Biol.*, PMID: 38321148) did a very deep dive into citrullination sites identified from HL60 leukemia cells differentiated into neutrophil-like cells. There, they also used a calcium ionophore to induce citrullination, used a neutrophil-like system, and the study was in human cells. Considering these similarities in both aim of the study and model systems, at a minimum, it would be prudent for the authors to compare their data (as described in Figure 4) to the citrullinome published by Rebak et al., as there should be a significant overlap (>50% based on a very quick Venn diagram), and it would be relevant to discuss any similarities or differences. The comparison could be done globally, like the authors did in Figure 3. More specifically, it would be interesting to compare histone citrullination (described by the authors in 4E, and in part listed by Rebak et al. in their Table 1), as from a quick glance there appears to be a strong overlap there as well.

In relation to the point above, the authors should include comparison to Rebak et al. in their discussion. Some of the points currently in the discussion, e.g. specific citrullinations on histones, were described by Rebak et al. and should as such be referenced. Of course, the authors can find a considerable number of citrullination sites from relatively short (27 – 80 min) single-shot runs, whereas Rebak et al. used over a day of gradient time per (46-fraction) replicate – a distinction that would certainly be worth mentioning in the context of scalability.

For the data table accompanying Figure 4 (i.e. Supplementary Table 9), it would be beneficial to have a list of citrullination sites that was identified (i.e. the “combined_site” that is present in Supplementary Table 8), as the current information is quite technical and verbose and only provided at the PSM level. Providing a more concise and cleaner table containing the sites (i.e. UniProt ID, gene name, citrulline position within the UniProt sequence, modified peptide, optionally a sequence window centered around the citrulline) would increase the resource value of the manuscript by making this information more accessible to a broader audience.

Same as above, but for Supplementary Table 6.

Unless I am mistaken, the methods section does not detail how the mouse RAW data was searched, nor where the library originated from.

In figure 4A the authors report their DDA gradient as 22 min. This likely relates to the active gradient, or the time where peptides are eluting. The actual gradient length (based on investigating RAW files) is 27 min, so the figure should be amended to either state “22 min active gradient” or “27 min”.

Reviewer #2

(Remarks to the Author)

In this manuscript, Gonzalez et al. describes the development of a high-throughput chemical workflow for global citrullinome profiling, which demonstrates a >10-fold increase in the detection of citrullinated peptides compared to other methods. This workflow increases citrullinome coverage in the analysis of tissues and, when applied to the study of NETs, reveals new dynamics of protein citrullination involved in remodeling the neutrophil structure during NETs formation. The manuscript is technically rigorous and the data and conclusions are not overstated. While the manuscript is better suited to a technical report in a biochemical journal, the study of citrullination dynamics during NET formation makes it highly innovative. Nevertheless, the study of NETs is also a major caveat of the manuscript.

1. In the manuscript, NETs were induced with ionomycin, a calcium ionophore that hyperactivates PADs, resulting in global hypercitrullination. Ionomycin is obtained from *Streptomyces conglobatus*, which is not pathogenic to humans. Importantly, among the large variety of mechanisms associated with the induction of NETs in humans, none has been linked to calcium ionophores. The problem is that hypercitrullination induced by ionomycin targets almost everything, including proteins and citrullination sites that are unlikely to be important for the formation of NETs. Under these conditions, it is impossible to distinguish between citrullinated proteins and citrullination sites that are directly involved in the process of NETs from those that are citrullinated bystanders. Thus, all ontology pathways and proteins linked to citrullination-induced NETs formation described in the manuscript could be artifacts due to the prominent effect of ionomycin. For instance, despite the general dogma, there is rigorous data from the Zychlinsky lab showing that citrullination is not required for the induction of NETs (PMID: 28574339). Moreover, MS analysis of NETs induced with RNP immune complexes (PMID: 28649674) only detected few citrullinated proteins (no more than 10) compared to the hundreds described in this manuscript. Certainly, the quality of the MS analysis in PMID: 28649674 is terrible, but not even citrullinated histones, which are easily detected by WB, were found in the NETs citrullinome. Thus, there is concern that the citrullination dynamics described in the manuscript may not be biologically relevant for NETs.

2. Detecting citrullinated proteins in ionomycin activated neutrophils is not a challenge. It can be done using any MS approach. The true test for the workflow described in the manuscript is to determine the citrullination dynamics during biologically significant NET formation. Over the last 20 years, the number of conditions linked to NETs has grown exponentially. Some include IL-8, LPS, *Candida albicans*, Group B *Streptococcus*, toxin-free *S. aureus*, RNP immune complexes (PMID: 28574339, 28649674, 15001782, 17210947) and many others. There are numerous options to consider. If the authors want to use the study of NETs to demonstrate the importance of their novel approach, the manuscript must include at least two examples of citrullination dynamics in NETs induced by any physiological or pathologic stimuli relevant to humans. Importantly, some bacteria secrete pore-forming toxins (e.g., *S. aureus*) that can induce necrosis and citrullination (PMID: 17210947), which could be confused with NETs. Contamination with pore-forming toxins should be avoided.

Reviewer #3

(Remarks to the Author)

Lee and coworkers reported a streamlined chemical proteomics workflow to study protein citrullination, an important protein post-translational modification that remains underexplored due to the lack of good methodology for proteome-wide profiling. The authors utilized commercially available 4-azido-phenyl glyoxal as a probe to derivatize citrulline in biological samples. This installed azido group onto citrullinated proteins, allowing click reaction to incorporate a cleavable biotin group for enrichment and subsequent release for MS analysis. They thoroughly investigated the MS results, including the fragmentation pattern of the derivatized peptides as well as the sensitivity and reproducibility of the detection of citrullinated sites and proteins. They successfully utilized their workflow to study protein citrullination in peptides, cell lysates, brain tissues and primary neutrophil cells, and identified previously uncharacterized citrullination sites on proteins such as on linker histone H1 and lamin B1.

The manuscript is well-written and is a pleasure to read. All the data have been well analyzed. Although the reagents used in their chemical proteomics workflow have been reported and the chemistry itself is not new, this is the first time protein citrullination has been profiled using both a clickable probe and a cleavable linker. This "clickable-cleavable" platform enabled a significant increase in the number of citrullinated peptide detection as compared to the reported workflow (Anal. Chem. 2022, 94, 17895). The authors also demonstrated the applicability of their workflow for studying protein citrullination in a variety of biological samples. This work should be of great interest to a broad readership, including those working on protein post-translational modifications, cell biology, chemical proteomics and mass spectrometry. Yet, there are some concerns that should be addressed prior to publication:

1. Selectivity of the probe derivatization of citrullination site

The peptides were reacted with a high concentration of APG probe (400 mM; from the Method section) in 50% TFA at 50 °C for 3 hours. The reaction condition is quite harsh. The authors should perform an experiment using control and probe-treated biological samples such as cell lysates, and then analyze probe selectivity and any undesired off-target modifications by open search.

2. Any changes of other protein PTMs under the strongly acidic conditions used for derivatization of citrullination site

The harsh reaction condition for probe derivatization may also remove/introduce other protein modifications. The authors discussed the potential PTM crosstalk between citrullination and histone methylation/acetylation (line 242). While it is interesting, the authors should perform experiments (e.g. proteomics for phosphorylation, acetylation and methylation) on control and probe-treated biological samples to study any changes in these PTMs. This can provide important information to the readers about the possibility of using their workflow together with proteomics platforms for other PTMs.

3. Open search to confirm selectivity of probe modification in the brain tissues and neutrophil samples

Glyoxal compounds are also known to react with arginine (but under basic condition. e.g. ACS Omega 2018, 3, 14229-14235). The authors should perform an open search on their MS data from more complex biological samples (brain tissues and neutrophils) to confirm that there are no modifications on arginine.

4. Profiling downregulated citrullination using their workflow

Although "profiling protein citrullination dynamics" is on the paper title, the authors have not investigated real biological samples with downregulation of citrullination. It is understandable that protein citrullination levels are normally low, so more protein input may be required for experiments on samples with downregulated citrullination. Yet, this is important to demonstrate the feasibility of their workflow to detect protein citrullination below the basal level and disclose the experimental details (particularly the protein input). The authors should perform experiments on samples treated with PAD4 inhibitors or with PAD4 knockdown.

Minor:

- Extended Data Fig.1: The chemical structure of dde-PEG-biotin alkyne is not correct

Version 1:

Reviewer comments:

Reviewer #1

(Remarks to the Author)

The authors have performed a very thorough set of follow-up experiments and validations, and they have addressed all my questions and comments. I highly recommend their manuscript for publication.

Reviewer #2

(Remarks to the Author)

The comments were properly addressed with rigorous data. I only have few minor comments.

1. Page 11, lane 25: The authors should consider the alternative that citrullination of ACTB, KRT1 and RPS6 is linked to *C. albicans* but not to ionomycin. If these proteins are constitutively modified, as mentioned by the authors, they should also be found in ionomycin-activated cells. The interaction and pathways activated during phagocytosis of *C. albicans* by neutrophils are very different to the simple activation of neutrophils by ionomycin. One possibility is that *C. albicans* induces changes in the neutrophil, not induced by ionomycin, that exposes ACTB, KRT1 and RPS6 to citrullination. Indeed, it is expected that different stimuli will generate different citrullinomes in neutrophils. Whether citrullination of ACTB, KRT1 and RPS6 is important for *C. albicans*-induced NETs or for some other function is unclear but should not be underestimated just because these are absent in neutrophils activated with ionomycin, which is an artificial stimulus.

2. Fig. 5c and d. I am not sure that the term "up-regulated" is adequate to describe proteins citrullinated by ionomycin. The assay addresses citrullination, not protein expression. Please consider using: ionomycin-induced citrullinome, citrullinated proteins induced by ionomycin, or any other alternative.

3. It is important to mention that RF-immune complexes failed to induce NETs. This information can be included anywhere, in results section or in methods. Indeed, it is consistent with previous work by Zychlinsky that failed to induce NETs using RNP-immune complexes (Elife. 2022 Oct 25;11:e68283. doi: 10.7554/eLife.68283. PMID: 36282064).

Reviewer #3

(Remarks to the Author)

The authors have convincingly addressed my comments and incorporated new data to investigate their probe labeling of citrullination sites vs arginine. They also successfully demonstrate the profiling of downregulated citrullination in primary human neutrophils treated with a PAD4 inhibitor.

I believe this revised manuscript is now ready for publication.

Point-by-point response

We thank the reviewers for their time and constructive feedback. In response, we have extensively revised the manuscript and performed additional experiments and analyses to address all major concerns. A clean and tracked-changes version of the revised manuscript is provided. Point-by-point responses are detailed below (page numbers refer to the clean version). Major revisions and new data include:

1. Validation with Pathogen-Induced Citrullinome in Neutrophils (New Fig. 5; Results page 11, line 3): We profiled primary human neutrophils stimulated with a fungal pathogen *Candida albicans*, identifying a conserved “core citrullinome” that substantially overlaps with the ionomycin-induced dataset (Response to Reviewer #2).
2. Quantification of Citrullination Down-regulation (New Extended Data Fig. 5; Results page 10, line 14): Using the PAD4 inhibitor GSK484, we demonstrate dose-dependent suppression of citrullination, strengthening the method’s capacity to detect citrullination dynamics (Response to Reviewer #3).
3. Search Engine Benchmarking and Localization Assessment (Extended Data Fig. 6): We compared MSFragger (hybrid and labile mode), MaxQuant, and Sage, highlighting sensitivity gains with MSFragger. We also verified site localization using PTMProphet within the FragPipe pipeline (Response to Reviewer #1).
4. Chemical Selectivity and Side-Reaction Characterization (Discussion page 14, lines 5–13): We directly addressed the +212 Da arginine derivatization, demonstrating that it is mass-resolvable from +213 Da derivatization, dose-independent, and computationally filtered (Response to Reviewer #3).
5. Expanded Discussion and Limitations: The Discussion has been updated to incorporate new data and address limitations raised by reviewers, including comments on selectivity, PTM compatibility, and computational aspects.

We are confident that all major issues have been thoroughly addressed, and we believe the revised manuscript is now suitable for publication.

Reviewer comments

Reviewer #1 (Remarks to the Author):

General

Meelker González et al. present a proteomics project in which they aim to enable the systemic study of citrullinated peptides via mass spectrometry, with the focus of specifically enriching this modification that has otherwise mainly been studied in a non-enriched (total proteome) context. Such an approach would allow the mass spectrometer to spend more time resolving citrullinated peptides, rather than irrelevant background peptides. In turn, this would enable the study of citrullination in more biologically relevant contexts, and at a scale that would be more affordable to many labs.

Current enrichment approaches for citrullinated peptides often lack specificity, are chemically quite harsh, or result in a purified sample that is not suitable for MS analysis. The authors

sought to work around this via a two-step derivatization and enrichment strategy, ultimately leaving a 213 Da mass remnant for each citrulline. They first validate this strategy, benchmark it in a setting with synthetic and spiked-in hyper-citrullinated HeLa, and finally test out the method in mouse brain as well as human primary neutrophils treated with a gradient of ionomycin.

The manuscript is well written and logically structured. The authors clearly explain the methodology and prove its efficacy in a variety of settings, giving equal importance to technical validation and application of their methodology to relevant biology. The strategy would be applicable by other labs, and potentially could be further refined in the future to enhance systems-wide profiling of citrullination by MS.

We appreciate the reviewer's thoughtful summary and positive overall assessment of our work.

I have a few queries regarding some of the MS data processing and quality-control throughout the manuscript. Overall, if these points are adequately addressed, I would recommend this manuscript for publication in Nature Communications.

Specific points

The authors define and search their mass remnants as “first derivative (Δ mass 158.0116), and the 509 final derivative of arginine (Δ mass 213.0538)”. In addition to mass deltas, the authors should provide the atomic composition of the derivatives relative to arginine, for the sake of clarity and so that this modification may be defined in other search engines. If I am correct, the atomic composition of the +213.05 would be C11 H7 N3 O2?

The reviewer is correct. We have added the atomic composition (C11 H7 N3 O2) to the Methods section of the revised manuscript (page 21, lines 22-23) for clarity and reproducibility across search engines.

As the authors are using data-dependent acquisition and are trying to pinpoint the location of citrullination within the peptides, correct localization is of high importance. The authors also mention the importance of localization throughout the manuscript, and in the methods describe that “Localize Mass Shift (LOS) was enabled” within the hybrid search of MSFragger. How exactly does the localization in MSFragger work? I was able to find a localization probability in some of the MSFragger output, which seems to either be “0” or “1”. Why are there no probabilities in between? While localizing a relatively large (213 Da) and stable PTM should not be too challenging, it would be beneficial if the authors could make an estimation/visualization of the distribution of localization confidence for the citrullinated peptides, especially since they (rightfully) choose to exclude peptide C-terminal citrullination.

We thank the reviewer for raising this important point. Because localization strategies differ fundamentally between search engines, we clarify and address this in two steps:

1. Clarification of “0” vs “1” (Deterministic Localization): Unlike MaxQuant, which uses a probabilistic algorithm (PTM Score) to report continuous probabilities (0–1), MSFragger's default localization is deterministic and score-based. When Localize Mass Shift (LOS) is enabled, MSFragger evaluates every possible residue position for the modification and calculates a Hyperscore.
 - 1 (Uniquely Localized): A single site has the highest score.
 - 0 (Ambiguous): There is a tie between sites or insufficient evidence to distinguish them.

Therefore, the field labeled “localization probability” in the standard FragPipe output is in fact a binary localization flag, not a probability. This explains why the reviewer observed only 0s and 1s.

2. Probabilistic Validation (PTMProphet): To provide the continuous confidence distribution requested by the reviewer, we re-analyzed representative biological datasets (HeLa, mouse brain, and neutrophils) using PTMProphet in Fragpipe (PMID: 31290668). This tool applies a Bayesian model to MSFragger results to report per-site probabilities with global false localization rate (FLR) control.

As shown below, all citrullinated peptides exceeded the default confidence threshold (>0.5), and the vast majority (>90%) showed localization probabilities >0.9, with medians between 0.96 and 1.00. This confirms that even though the default output is binary, the underlying site assignments are highly confident.

The authors should provide an estimate or distribution of overall spectral quality for all complex datasets (figures 2, 3, 4). The Astral data (Figure 4) seems to be of high spectral quality, via manual inspection of the RAW data and a quick search via MaxQuant. However, based on a manual investigation of some of the MS/MS spectra within e.g. the RAW data generated for Figure 3 (YIG_314_L001_105_002_RMG, YIG_314_L001_105_005_RMG, and YIG_314_L001_105_008_RMG) using a Fusion Lumos, the quality of these MS/MS spectra is overall not very high. Analyzing these files with MaxQuant yields a relatively low MS/MS identification rate (~7%), with average spectral scores of ~83 for all identified peptides, and only approximately 30% of peptides exceed an Andromeda score of 100 (which is generally a minimum for a spectrum that would be manually annotatable). For the +213 Da modified peptides, which appear to score lower than unmodified background peptides, only about 10% of spectra appear to be in excess of 100 Andromeda score. Based on a (quick and perhaps sub-optimal) MaxQuant search of the 3 mouse RAW files I was able to find ~100 unique +213 Da sites, whereas the authors find ~400 using MSFragger (in both cases excluding peptide C-terminal site assignments). Perhaps MSFragger (in combination with post-processing using PeptideProphet) is more efficient at identifying this modification compared to MaxQuant, but still the difference seems relatively large. Using an alternative search engine on the data and being able to provide numbers in the same ballpark as MSFragger, which could be included as a technical note, would strengthen the claims made by the authors.

We have addressed the questions regarding (1) spectral quality and (2) search engine sensitivity below.

1. Spectral Quality Assessment: We agree that spectral quality is key for modified peptides. We plotted Hyperscore distributions (MSFragger’s main scoring metric) for both derivatized (+213 Da) and unmodified peptides across three datasets (See below). We observed that derivatized peptides scored slightly lower, particularly in the mouse brain dataset. We attribute this to two specific factors: i) Low endogenous citrullination in unstimulated brain, and ii) Instrument differences: the Fusion Lumos (brain) produces lower MS2 quality than the Astral (neutrophils). Despite these lower scores, we observed a citrullination motif strongly consistent with that found in the human brain (Fig. 3f), providing orthogonal support for correct identifications.

2. MSFragger vs. MaxQuant Sensitivity: The reviewer notes a discrepancy in identifications (~400 vs ~100) in the mouse brain dataset. We believe this reflects analytical sensitivity rather than specificity. MaxQuant does not currently support “labile” fragment searching. The derivatized citrulline moiety (+213 Da) often undergoes neutral loss during fragmentation. MaxQuant might miss the resulting diagnostic ions, whereas MSFragger’s hybrid offset + labile mode is specifically configured to detect these “remainder” ions. To demonstrate that this is not an artifact, we benchmarked MSFragger, MaxQuant, and Sage (PMID: 37819886) in a controlled HeLa citrullination dilution series (Dataset used in Fig. 2e). MSFragger consistently reported more PSMs across all input levels (See below). We conclude that MSFragger offers improved sensitivity without sacrificing specificity. We have included this technical point in the Discussion (page 12, lines 11-18) and Extended Fig. 6.

Extended Data Fig. 6

In relation to the two points above, and since we are looking at DDA data for modified peptide sequences, the authors should provide fully annotated MS/MS spectra for all citrullinated

peptides identified in the manuscript, stratified per experiment. Currently there is only one annotated spectrum in Figure 1F, which is missing virtually the entire b-ion series. Providing annotated spectra (e.g. online via MS-Viewer, or batch-exported from any DDA search engine) would increase transparency and simultaneously demonstrate sufficient spectral quality to facilitate accurate identification and localization of citrullination within the peptides. Without the ability to manually inspect annotated spectra, and perhaps also an orthogonal data search using a different software, it would be difficult to ascertain the spectral quality and rule out over-interpretation of the data.

We fully support the request for transparency and manual inspection. To view any identified peptides in our results, we propose to access them interactively via Proteomics Data Viewer (PDV, PMID: 30169737), which allows interactive inspection of spectra.

Instructions for usage:

- Find scan number and file name in Supplementary Tables.
- Open corresponding .mzML and .pepXML files from MassIVE in PDV (available at <https://wenbostar.github.io/PDV>).
- Use PDV's GUI to visualize b/y-ion series and neutral losses.

This platform provides greater transparency than static figures and allows readers to explore fragmentation patterns interactively (see below for example).

The authors derivatize citrullination, and search for the 213 Da derivative on Arginine residues. Is it possible for this derivative to exist on other amino acid residues? While this is to an extent addressed in Figure 1, it would be important for the authors to perform an in silico entrapment experiment, by searching a subset of the data generated from complex samples (e.g. from Figure 2, 3, or 4) and allowing the 213 Da mass remnant to be assigned to e.g. glutamine and asparagine residues in addition to arginine residues (or ideally allow it on any residue to perform a true localization experiment, but this may be computationally challenging). This would give an indication of the overall false discovery rate of the author's methodology when

applied to complex samples – either because of non-specific chemical derivatization (i.e. in vitro), or in relation to the accuracy of the computational localization (i.e. in silico).

We address this concern with chemical and computational evidence.

- 1) Chemical selectivity of derivatization: APG selectively reacts with citrulline via the ureido group under strongly acidic conditions (50% TFA) and with arginine under basic conditions. No plausible chemical mechanism exists for this reaction with canonical amino acid side chains such as Q, N, or D. We performed open searches across HeLa, mouse brain, and neutrophils (See Reviewer#3 Comment #2). The only recurring delta masses observed were +212 Da (Arg side-reaction) and +213 Da (Cit). No other residues consistently showed masses in this range.
- 2) Computational localization: We use PTMProphet to model per-site localization with global FLR control. The majority of modified sites have high probabilities (>0.9), with no enrichment of off-residue assignments. Refer to response point #2 above for the full localization distribution.

While the reviewer's proposed entrapment search (e.g. allowing +213 Da on Q/N) is a useful theoretical control, our open search and site-localization results already rule out consistent off-target derivatization or misassignment. These lines of evidence collectively suggest that the 213 Da modification is both chemically selective for citrulline and computationally well-localized in our datasets.

In their final experiment (Figure 4), the authors enrich and identify citrullination from primary human neutrophils stimulated with ionomycin, which is a calcium ionophore. Although briefly mentioned in the introduction, a recent landmark publication by Rebak et al. (published in *Nat. Struct. Mol. Biol.*, PMID: 38321148) did a very deep dive into citrullination sites identified from HL60 leukemia cells differentiated into neutrophil-like cells. There, they also used a calcium ionophore to induce citrullination, used a neutrophil-like system, and the study was in human cells. Considering these similarities in both aim of the study and model systems, at a minimum, it would be prudent for the authors to compare their data (as described in Figure 4) to the citrullinome published by Rebak et al., as there should be a significant overlap (>50% based on a very quick Venn diagram), and it would be relevant to discuss any similarities or differences. The comparison could be done globally, like the authors did in Figure 3. More specifically, it would be interesting to compare histone citrullination (described by the authors in 4E, and in part listed by Rebak et al. in their Table 1), as from a quick glance there appears to be a strong overlap there as well.

We appreciate the reviewer's suggestion to compare our dataset with the recent study by Rebak *et al.*, a major profiling effort in neutrophil-like cells. We have now included this comparison in Extended Data Fig. 7 and updated the Discussion (page 13, lines 19–26), with appropriate referencing throughout.

Despite substantial methodological differences—cell type (culture cell-derived vs. primary neutrophils), MS runtime (~1 day vs. 22 min per sample), and sample prep (46-fraction vs. single-shot)—we observed ~50% overlap in citrullination sites, including histones, supporting the robustness of our approach (see below).

While Rebak *et al.* generated a comprehensive “citrullinome atlas” with broad site coverage, our study emphasizes functional profiling. Using PAD activation and inhibition in a dose-dependent design, we identified regulated and PAD-responsive sites, distinguishing functional

citrullination from basal modifications. This added regulatory dimension, along with the scalability of our workflow, reinforces its utility for studying citrullination dynamics in physiological contexts. While a deeper comparison of e.g. histone citrullination is certainly of interest, we note that our analysis focuses on the subset of functionally regulated histone sites, which were not statistically defined in the Rebak study.

Therefore, this distinction between total coverage and regulatory insight highlights the value of our workflow for dissecting citrullination dynamics in physiologically relevant settings for larger studies.

Extended Data Fig. 7

In relation to the point above, the authors should include comparison to Rebak et al. in their discussion. Some of the points currently in the discussion, e.g. specific citrullinations on histones, were described by Rebak et al. and should as such be referenced. Of course, the authors can find a considerable number of citrullination sites from relatively short (27 – 80 min) single-shot runs, whereas Rebak et al. used over a day of gradient time per (46-fraction) replicate – a distinction that would certainly be worth mentioning in the context of scalability.

We have addressed this point in the response above and have updated the Discussion accordingly (page 13, lines 19–26).

For the data table accompanying Figure 4 (i.e. Supplementary Table 9), it would be beneficial to have a list of citrullination sites that was identified (i.e. the “combined_site” that is present in Supplementary Table 8), as the current information is quite technical and verbose and only provided at the PSM level. Providing a more concise and cleaner table containing the sites (i.e. UniProt ID, gene name, citrulline position within the UniProt sequence, modified peptide, optionally a sequence window centered around the citrulline) would increase the resource value of the manuscript by making this information more accessible to a broader audience.

Done. We have added a “combined_site” sheet to Supplementary Tables 6 and 9, and new Supplementary Tables 10 and 11.

Same as above, but for Supplementary Table 6.

Done.

Unless I am mistaken, the methods section does not detail how the mouse RAW data was searched, nor where the library originated from.

Thank you for flagging this omission. We have added a detailed description in the Methods (page 21-22). The search parameters were consistent with those used for the HeLa dilution series.

In figure 4A the authors report their DDA gradient as 22 min. This likely relates to the active gradient, or the time where peptides are eluting. The actual gradient length (based on investigating RAW files) is 27 min, so the figure should be amended to either state “22 min active gradient” or “27 min”.

Correct. We have now revised Figure 4 and Methods (page 21, line 1) to state “22 min active gradient”.

Reviewer #2 (Remarks to the Author):

In this manuscript, Gonzalez et al. describes the development of a high-throughput chemical workflow for global citrullinome profiling, which demonstrates a >10-fold increase in the detection of citrullinated peptides compared to other methods. This workflow increases citrullinome coverage in the analysis of tissues and, when applied to the study of NETs, reveals new dynamics of protein citrullination involved in remodeling the neutrophil structure during NETs formation. The manuscript is technically rigorous and the data and conclusions are not overstated. While the manuscript is better suited to a technical report in a biochemical journal, the study of citrullination dynamics during NET formation makes it highly innovative. Nevertheless, the study of NETs is also a major caveat of the manuscript.

We thank the reviewer for recognizing the technical rigor and innovation of our study. We respond below to the specific points raised.

1. In the manuscript, NETs were induced with ionomycin, a calcium ionophore that hyperactivates PADs, resulting in global hypercitrullination. Ionomycin is obtained from *Streptomyces conglobatus*, which is not pathogenic to humans. Importantly, among the large variety of mechanisms associated with the induction of NETs in humans, none has been linked to calcium ionophores. The problem is that hypercitrullination induced by ionomycin targets almost everything, including proteins and citrullination sites that are unlikely to be important for the formation of NETs. Under these conditions, it is impossible to distinguish between citrullinated proteins and citrullination sites that are directly involved in the process of NETs from those that are citrullinated bystanders. Thus, all ontology pathways and proteins linked to citrullination-induced NETs formation described in the manuscript could be artifacts due to the prominent effect of ionomycin. For instance, despite the general dogma, there is rigorous data from the Zychlinsky lab showing that citrullination is not required for the induction of NETs (PMID: 28574339). Moreover, MS analysis of NETs induced with RNP immune complexes (PMID: 28649674) only detected few citrullinated proteins (no more than 10) compared to the hundreds described in this manuscript. Certainly, the quality of the MS analysis in PMID: 28649674 is terrible, but not even citrullinated histones, which are easily detected by WB, were found in the NETs citrullinome. Thus, there is concern that the citrullination dynamics described in the manuscript may not be biologically relevant for NETs.

We fully agree that not all NET pathways involve calcium/PAD activation, nor are all citrullinated proteins detected under PAD stimulation functionally relevant. Identifying biologically meaningful sites is a key goal of citrullinome profiling.

Ionomycin was selected as it is a well-established reagent for strictly modeling PAD-dependent NETosis. While not pathogen-derived, it induces a distinct, calcium-driven pathway separate

from oxidative mechanisms, offering controlled and reproducible PAD activation in primary neutrophils (Kenny *et al.*, 2017, PMID: 28574339; Wang *et al.*, 2009, PMID: 19153223).

To ensure biological relevance beyond broad hypercitrullination, we designed a dose–response experiment that:

1. Quantifies up-regulated citrullination events during dose-dependent ionomycin-induced NET formation instead of single high dose (Fig. 4),
2. Dissects basal vs. PAD4-mediated sites through inhibition with GSK484 (new Extended Data Fig. 5),
3. Enables pEC50 estimation of down-regulated sites and defines a core citrullinome (384 activation- and 190 inhibition-responsive sites; Fig 4 and new Extended Data Fig. 5).

These results revealed that only a subset of detected citrullination sites is strongly dose-responsive and PAD4-mediated, suggesting their functional relevance. The remaining sites likely represent baseline modifications, background reactivity, or are mediated by PAD2 isozyme. Furthermore, regulated sites were enriched in solvent-exposed and disordered regions, consistent with known PAD substrate preferences, and showed motif features associated with PAD activity.

Finally, gene ontology (GO) analysis was restricted to regulated citrullination sites, not the full list of detected modifications. The enriched pathways—chromatin remodeling, cytoskeletal reorganization, and organelle structure—are highly consistent with known cellular changes during NET formation and PAD activation with more site-specific resolution.

Thus, we believe our ionomycin-based dose-dependent profiling provides biologically informative and selective insight into PAD-driven citrullination events relevant to NETosis.

2. Detecting citrullinated proteins in ionomycin activated neutrophils is not a challenge. It can be done using any MS approach. The true test for the workflow described in the manuscript is to determine the citrullination dynamics during biologically significant NET formation. Over the last 20 years, the number of conditions linked to NETs has grown exponentially. Some include IL-8, LPS, *Candida albicans*, Group B *Streptococcus*, toxin-free *S. aureus*, RNP immune complexes (PMID: 28574339, 28649674, 15001782, 17210947) and many others. There are numerous options to consider. If the authors want to use the study of NETs to demonstrate the importance of their novel approach, the manuscript must include at least two examples of citrullination dynamics in NETs induced by any physiological or pathologic stimuli relevant to humans. Importantly, some bacteria secrete pore-forming toxins (e.g., *S. aureus*) that can induce necrosis and citrullination (PMID: 17210947), which could be confused with NETs. Contamination with pore-forming toxins should be avoided.

We agree and have addressed this by profiling primary neutrophils stimulated with heat-killed *Candida albicans* (HKCA), a clinically relevant pathogen shown to induce PAD4-dependent NETosis (PMID: 28574339, 25622091). This experiment is now detailed in new Figure 5 (below) and described in the revised Results (page 11, lines 3-30). Briefly, we found that:

1. In 4 biological replicates, we identified 268 citrullination sites in HKCA-stimulated neutrophils, vs. ~40 in serum-treated controls.
2. ~200 sites were consistently observed in ≥ 2 replicates, and ~87% overlapped with ionomycin-regulated sites (Fig. 5c).

- GO analysis revealed strong enrichment in chromatin organization, cytoskeletal remodeling, and organelle structure—highly consistent with the ionomycin-regulated citrullinome (Fig. 5e).
- Shared targets included key NET components like histones, lamin B1, and cytoskeletal proteins, defining a reproducible “core citrullinome” conserved across NET stimuli (Fig. 5f).

We also attempted stimulation with RF-immune complexes (PMID: 28574339), but were unable to identify robust and reproducible activation conditions. Nonetheless, the substantial overlap and biological concordance between HKCA- and ionomycin-induced citrullination profiles confirm that our workflow captures a physiologically meaningful citrullination signature relevant to NET formation.

Figure 5

editorial note: figure 5, panel a, Created in BioRender. Lee, C. (2025) BioRender.com/nkia0e

Reviewer #3 (Remarks to the Author):

Lee and coworkers reported a streamlined chemical proteomics workflow to study protein citrullination, an important protein post-translational modification that remains underexplored due to the lack of good methodology for proteome-wide profiling. The authors utilized commercially available 4-azido-phenyl glyoxal as a probe to derivatize citrulline in biological samples. This installed azido group onto citrullinated proteins, allowing click reaction to incorporate a cleavable biotin group for enrichment and subsequent release for MS analysis. They thoroughly investigated the MS results, including the fragmentation pattern of the derivatized peptides as well as the sensitivity and reproducibility of the detection of citrullinated sites and proteins. They successfully utilized their workflow to study protein citrullination in peptides, cell lysates, brain tissues and primary neutrophil cells, and identified previously uncharacterized citrullination sites on proteins such as on linker histone H1 and lamin B1.

The manuscript is well-written and is a pleasure to read. All the data have been well analyzed. Although the reagents used in their chemical proteomics workflow have been reported and the

chemistry itself is not new, this is the first time protein citrullination has been profiled using both a clickable probe and a cleavable linker. This “clickable-cleavable” platform enabled a significant increase in the number of citrullinated peptide detection as compared to the reported workflow (Anal. Chem. 2022, 94, 17895). The authors also demonstrated the applicability of their workflow for studying protein citrullination in a variety of biological samples. This work should be of great interest to a broad readership, including those working on protein post-translational modifications, cell biology, chemical proteomics and mass spectrometry.

We thank the reviewer for their thoughtful and encouraging assessment of our work.

Yet, there are some concerns that should be addressed prior to publication:

1. Selectivity of the probe derivatization of citrullination site

The peptides were reacted with a high concentration of APG probe (400 mM; from the Method section) in 50% TFA at 50 °C for 3 hours. The reaction condition is quite harsh. The authors should perform an experiment using control and probe-treated biological samples such as cell lysates, and then analyze probe selectivity and any undesired off-target modifications by open search.

To address this concern, we performed an open search comparing untreated and probe-treated HeLa lysates. As shown below, the vast majority of identified peptides carried no modification. In the probe-treated samples, we observed ~90 PSMs with delta masses in the 200–250 Da range. These correspond primarily to endogenous citrullinated proteins and a minor +212 Da arginine side-reaction (discussed in detail in Comment #3), rather than broad off-target derivatization.

Importantly, we did not observe any other recurring or enriched mass shifts outside this expected window. Together, these data indicate that undesired side reactions in complex lysate backgrounds are limited and that APG derivatization is largely selective for citrulline under our acidic labeling conditions.

2. Any changes of other protein PTMs under the strongly acidic conditions used for derivatization of citrullination site

The harsh reaction condition for probe derivatization may also remove/introduce other protein modifications. The authors discussed the potential PTM crosstalk between citrullination and histone methylation/acetylation (line 242). While it is interesting, the authors should perform experiments (e.g. proteomics for phosphorylation, acetylation and methylation) on control and probe-treated biological samples to study any changes in these PTMs. This can provide important information to the readers about the possibility of using their workflow together with proteomics platforms for other PTMs.

We agree this is an important limitation. Our derivatization conditions (50% TFA, 50 °C, 3 hr) are optimized for citrulline reactivity, but may destabilize acid-labile PTMs such as phosphorylation and some forms of acetylation.

We did not pursue co-enrichment of multiple PTMs in this workflow. As noted in the revised Discussion (page 14, lines 19-24), we now recommend performing e.g. acetylation/methylation/phosphorylation analyses in parallel workflows rather than on derivatized samples when PTM crosstalk is a key question.

3. Open search to confirm selectivity of probe modification in the brain tissues and neutrophil samples

Glyoxal compounds are also known to react with arginine (but under basic condition. e.g. ACS Omega 2018, 3, 14229-14235). The authors should perform an open search on their MS data from more complex biological samples (brain tissues and neutrophils) to confirm that there are no modifications on arginine.

We performed open searches on mouse brain and neutrophil datasets to assess the selectivity of APG derivatization in complex biological samples. A subset of peptides with a +212.0700 Da modification on arginine residues was identified (see below). This likely reflects glyoxal-based reactivity toward arginine under basic pH, potentially during SCX cleanup, rather than a direct result of the acidic APG labeling step.

To further assess the quantitative impact of these events, we re-searched our *in vitro* HeLa dilution series with both variable modifications of +213.0538 Da (Cit-derivatized) and +212.0700 Da (Arg-derivatized). In these data (see below and Extended Fig. 8):

1. +213.0538 Da (Cit-derivatized) peptides showed a clear dose-dependent increase in identification count and intensity.
2. +212.0700 Da (Arg-derivatized) peptides remained constant or slightly decreased with increasing citrullination levels.
3. The majority of +212 Da PSMs were C-terminal arginine peptides, likely reflecting highly abundant tryptic peptides that are consistently present across samples.

- Non-C-terminal +212 Da sites showed stable intensity across dilutions, consistent with a background signal rather than biologically meaningful citrullination.

These observations support the conclusion that +212 Da events represent a non-regulated background artifact. Crucially, the mass difference between +212 and +213 Da is readily resolved by high-resolution MS, allowing clear distinction in our analyses.

To ensure analytical rigor, our data processing pipeline includes explicit filtering of +212 Da events by:

- Restricting site localization to +213.0538 Da for citrulline.
- Excluding C-terminally modified peptides.
- Validating consistent replicate or dose-dependent behavior.

We have now included this clarification in the revised Discussion section detailing these observations, so readers can apply similar precautions (page 14, lines 5-18).

Extended Data Fig. 8

4. Profiling downregulated citrullination using their workflow

Although “profiling protein citrullination dynamics” is on the paper title, the authors have not investigated real biological samples with downregulation of citrullination. It is understandable that protein citrullination levels are normally low, so more protein input may be required for experiments on samples with downregulated citrullination. Yet, this is important to demonstrate the feasibility of their workflow to detect protein citrullination below the basal level and disclose the experimental details (particularly the protein input). The authors should perform experiments on samples treated with PAD4 inhibitors or with PAD4 knockdown.

We agree that demonstrating the ability to quantify down-regulated citrullination is critical for validating dynamic profiling. Because basal citrullination is minimal in unstimulated cells, inhibiting an unstimulated system would not be informative. Thus, we assessed down-regulation in the context of PAD4 inhibition following ionomycin activation in primary human neutrophils.

Neutrophils were pre-treated with a dose range of the PAD4-selective inhibitor GSK484 (0.3–30 μ M) and then activated with ionomycin. This experiment revealed (see Extended Data Fig. 5 and Results section page 10, lines 14-33):

- Quantification of 1,568 citrullination sites across all conditions,
- A dose-dependent reduction in total citrullinated peptide abundance,

3. Identification of 190 significantly down-regulated sites, of which 103 overlapped with ionomycin-up-regulated targets—thereby defining a PAD4-responsive core citrullinome,
4. EC₅₀ values for inhibition were consistent across all regulated sites.

These findings confirm that our method can detect both increases and decreases in citrullination, providing site-specific resolution of PAD4 activity in physiologically relevant settings.

Minor:

- Extended Data Fig.1: The chemical structure of dde-PEG-biotin alkyne is not correct

Corrected. Thank you for catching this error.

Point-by-point response

Reviewer comments

Reviewer #2 (Remarks to the Author):

The comments were properly addressed with rigorous data. I only have few minor comments.

1. Page 11, lane 25: The authors should consider the alternative that citrullination of ACTB, KRT1 and RPS6 is linked to *C. albicans* but not to ionomycin. If these proteins are constitutively modified, as mentioned by the authors, they should also be found in ionomycin-activated cells. The interaction and pathways activated during phagocytosis of *C. albicans* by neutrophils are very different to the simple activation of neutrophils by ionomycin. One possibility is that *C. albicans* induces changes in the neutrophil, not induced by ionomycin, that exposes ACTB, KRT1 and RPS6 to citrullination. Indeed, it is expected that different stimuli will generate different citrullinomes in neutrophils. Whether citrullination of ACTB, KRT1 and RPS6 is important for *C. albicans*-induced NETs or for some other function is unclear but should not be underestimated just because these are absent in neutrophils activated with ionomycin, which is an artificial stimulus.

We thank the reviewer for this comment. We have re-verified our dataset and confirm that ACTB, KRT1, and RPS6 are indeed detected in the ionomycin-stimulated dataset. However, they do not show statistically significant up-regulation upon stimulation. This lack of regulation in the ionomycin model supports our interpretation that they likely represent constitutively modified basal proteins rather than NET-specific signaling events. We have revised the text to explicitly state that these proteins were detected but unregulated (see page.11, lines 25-26)

Revised text: "Interestingly, several highly citrullinated proteins in the HKCA condition—such as ACTB, KRT1, and RPS6—*while detected in the ionomycin dataset, were not significantly regulated*. These likely represent constitutively modified proteins present in basal neutrophils, rather than NET-specific citrullination events."

(Fig 5f: HKCA)

(Same sets of proteins in Ionomycin dataset)

2. Fig. 5c and d. I am not sure that the term “up-regulated” is adequate to describe proteins citrullinated by ionomycin. The assay addresses citrullination, not protein expression. Please consider using: ionomycin-induced citrullinome, citrullinated proteins induced by ionomycin, or any other alternative.

We understand the reviewer’s concern regarding the distinction between protein expression and PTM abundance. In this context, we used "up-regulated" to strictly denote statistically significant increases in citrullinated peptide abundance. To avoid ambiguity, we have revised the figure (Fig 5c/d) to refer to these as "significantly induced by ionomycin”

Figure 5c-d

3. It is important to mention that RF-immune complexes failed to induce NETs. This information can be included anywhere, in results section or in methods. Indeed, it is consistent with previous work by Zychlinsky that failed to induce NETs using RNP-immune complexes (Elife. 2022 Oct 25;11:e68283. doi: 10.7554/eLife.68283. PMID: 36282064).

We agree. We have added a statement to the Discussion explicitly noting that RF-immune complexes failed to induce robust NETosis in our hands, consistent with Zychlinsky et al. (eLife 2022). Please see page 13, lines 4-7.

Revised text: “Beyond calcium ionophore stimulation, we validated the physiological relevance of our findings using the fungal pathogen *Candida albicans*. **We also attempted to induce NETosis using Rheumatoid Factor (RF)-immune complexes; however, consistent with previous reports⁵⁹, this stimulus failed to trigger robust NET formation in our primary neutrophil model.** Importantly, while the total number of induced citrullination sites from *C. albicans* was lower than in the ionomycin-stimulated condition, we observed a high degree of overlap (~87%) between the ionomycin- and *C. albicans*-induced citrullinomes.